# PARAMETER-EFFICIENT LONG-TAILED RECOGNITION

## ABSTRACT

The "pre-training and fine-tuning" paradigm in addressing long-tailed recognition tasks has sparked significant interest since the emergence of large vision-language models like the contrastive language-image pre-training (CLIP). While previous studies have shown promise in adapting pre-trained models for these tasks, they often undesirably require extensive training epochs or additional training data to maintain good performance. In this paper, we propose PEL, a fine-tuning method that can effectively adapt pre-trained models to long-tailed recognition tasks in fewer than 20 epochs without the need for extra data. We first empirically find that commonly used fine-tuning methods, such as full fine-tuning and classifier fine-tuning, suffer from overfitting, resulting in performance deterioration on tail classes. To mitigate this issue, PEL introduces a small number of task-specific parameters by adopting the design of any existing parameter-efficient fine-tuning method. Additionally, to expedite convergence, PEL presents a novel semantic-aware classifier initialization technique derived from the CLIP textual encoder without adding any computational overhead. Our experimental results on four long-tailed datasets demonstrate that PEL consistently outperforms previous state-of-the-art approaches. The source code is available in the supplementary material.

## 1 INTRODUCTION

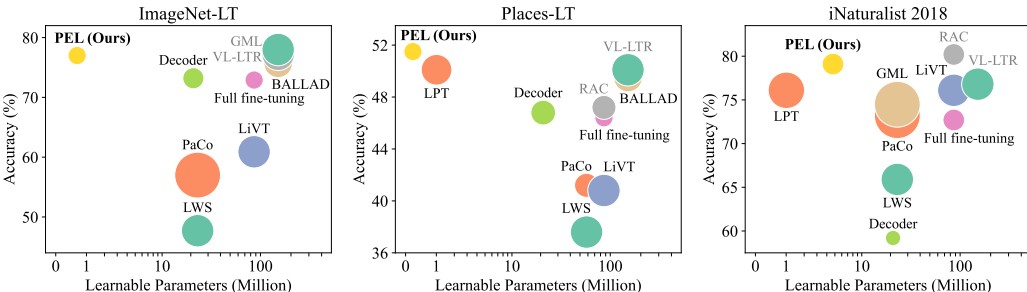

Figure 1: Comparison with prior state-of-the-art methods. The x-axis represents the number of learnable parameters, while the y-axis shows the test accuracy. **The size of each point corresponds to the number of training epochs**, with larger points indicating longer training times. Gray labels denote methods that incorporate external data. PEL consistently achieves higher performance with lower computational costs and is even comparable with methods that leverage external data.

Long-tailed recognition addresses the challenge of learning from highly imbalanced data, where a small set of classes (head classes) is well-represented in the training data, while other classes (tail classes) have only a limited number of training samples available. Given its widespread attention, numerous long-tailed recognition approaches have emerged to enhance generalization, particularly for tail classes. These approaches typically fall into three categories: 1) data manipulation (Zhou et al., 2020; Kang et al., 2020), 2) representation learning (Zhong et al., 2021; Cui et al., 2021), and 3) model output adjustment (Menon et al., 2021; Ren et al., 2020). While these existing methods have made substantial strides in improving classification accuracy, a significant gap still persists compared to models trained on class-balanced datasets.

Instead of training deep neural networks from scratch, recent results from BALLAD (Ma et al., 2021), RAC (Long et al., 2022), VL-LTR (Tian et al., 2022), and LPT (Dong et al., 2023) show that

properly fine-tuning pre-trained models can surprisingly improve the long-tailed recognition performance. For example, LPT fine-tunes the vision Transformer pre-trained on ImageNet, utilizing prompt tuning (Jia et al., 2022) via two-phrase training. VL-LTR adopts contrastive language-image pre-training (CLIP) (Radford et al., 2021) and incorporates additional image-text web data for fine-tuning. Nevertheless, the performance improvements come at the cost of 1) longer training epochs ($\approx 100$), 2) a two-staged procedure, and 3) an external training dataset ($size \approx 10^6$).

To overcome the aforementioned limitations, we introduce PEL, a novel and unified approach for fine-tuning pre-trained models in long-tailed recognition. We empirically discover that two prevalent fine-tuning methods, *i.e.,* full fine-tuning (which fine-tunes all network parameters) and classifier fine-tuning (which focuses solely on the classifier), both suffer from issues of overfitting. In contrast to previous efforts, this paper explores parameter-efficient fine-tuning (PEFT), where pre-trained parameters remain fixed while incorporating a limited number of task-specific learnable parameters. This approach preserves the discriminative capacity required for handling tail classes effectively. Moreover, our proposed framework is general, allowing for the integration of various PEFT methods such as VPT (Jia et al., 2022), LoRA (Hu et al., 2022), and Adapter (Houlsby et al., 2019).

Despite PEFT optimizing only a small set of parameters, training Transformer models on large datasets still incurs notable computational overhead. In pursuit of fast convergence, we introduce a simple yet effective classifier initialization method. Specifically, we harness textual features extracted from CLIP using the prompt template to initialize the classifier weights, drawing upon knowledge from the pre-trained CLIP model and leveraging semantic relationships among classes. In terms of model optimization, this equips PEL with a robust starting point for adapting pre-trained models, even in scenarios where only a small number of training samples are available.

The contributions of this paper are summarized as follows: **1)** We identify a critical limitation in commonly used fine-tuning methods, such as full fine-tuning and classifier fine-tuning, revealing their susceptibility to overfitting on tail classes. **2)** We introduce a unified parameter-efficient tuning framework, featuring a novel semantic-aware classifier initialization technique and a test-time ensemble method. **3)** Through comprehensive experiments, we demonstrate that our proposed method consistently outperforms the prior state-of-the-art on four benchmark datasets. **4)** Our method stands out as a one-staged approach, achieving convergence in fewer than 20 training epochs without requiring supplementary training data, which significantly improves its practicality.

## 2 RELATED WORKS

**Long-tailed recognition via training from scratch.** Conventional methods train convolutional neural network models like ResNet and ResNeXt on long-tailed datasets. Concerning the class imbalance, there are three main directions to improve the performance: 1) data manipulation (Zhou et al., 2020; Kang et al., 2020), 2) representation learning (Zhong et al., 2021; Cui et al., 2021), and 3) model output adjustment (Menon et al., 2021; Ren et al., 2020). Data manipulation typically includes designing re-sampling strategies, and data augmentations. Many works improve the performance by adopting two-stage training where the first stage learns representations and the second stage learns the classifier (Zhong et al., 2021; Wei & Gan, 2023). The adjustment of the model's outputs can be done during training by optimizing unbiased loss functions or after training. In contrast to the aforementioned works, this paper presents an end-to-end training framework that combines the advantages of multiple existing techniques through optimizing an unbiased loss function and ensembling the model outputs generated by several data augmentations during test time.

**Long-tailed recognition via fine-tuning pre-trained model.** Recent progresses utilize pre-trained Transformer models such as CLIP (Radford et al., 2021) and ViT (Dosovitskiy et al., 2021). Fine-tuning models pre-trained on large-scale datasets has emerged as an effective strategy to address class imbalance due to the strong representation learning capabilities (Ma et al., 2021; Long et al., 2022; Tian et al., 2022; Dong et al., 2023). However, it is important to note that these methods often require prolonged training iterations and, in some cases, rely on external training datasets to facilitate the learning. In contrast, our proposed approach exhibits the remarkable ability to achieve convergence in fewer than 20 epochs and does not need external data. Furthermore, our method is general, allowing for seamless integration with various parameter-efficient fine-tuning approaches.

## 3 PARAMETER-EFFICIENT LONG-TAILED RECOGNITION

### 3.1 ZERO-SHOT CLIP IS A STRONG REPRESENTATION LEARNER

**Preliminary.** Different from convolutional neural networks, the Transformer architecture (Vaswani et al., 2017; Dosovitskiy et al., 2021) is more simply designed while exhibiting remarkable capabilities. A Transformer model consists of an embedding layer and multiple Transformer blocks. Formally, it first divides an input image $x$ into $m$ patches $\{x_i^{\mathrm{p}}\}_{i=1}^m$. These patches are then embeded into sequences of $d$-dimensional vectors $E^0 = \mathrm{Embed}([x_1^{\mathrm{p}}; \cdots; x_m^{\mathrm{p}}]) \in \mathbb{R}^{m \times d}$. The input embeddings are subsequently passed through $L$ Transformer blocks $\{\Phi^l\}_{l=1}^L$ within the model:

$$X^l = \Phi^l(X^{l-1}). \text{ Specifically, } \begin{cases} \hat{X}^l = \mathrm{MSA}^{(l)}(\mathrm{LN}(X^{l-1})) + X^{l-1} \\ X^l = \mathrm{FFN}^{(l)}(\mathrm{LN}(\hat{X}^l)) + \hat{X}^l \end{cases} \tag{1}$$

$$\mathrm{MSA}^{(l)}(X) = \mathrm{Concat}_{h=1}^H \left( \mathrm{Softmax}\left( \frac{X W_Q^{l,h}(X W_K^{l,h})^\top}{\sqrt{d}} \right) X W_V^{l,h} \right) W_O^l \tag{2}$$

$$\mathrm{FFN}^{(l)}(X) = \mathrm{ReLU}(X W_1^l) W_2^l \tag{3}$$

Here, MSA denotes the multi-head self-attention and $H$ is the number of heads. FFN indicates the feed-forward network, and LN denotes layer normalization (Ba et al., 2016). $W_Q^{l,h}, W_K^{l,h}, W_V^{l,h} \in \mathbb{R}^{d \times \frac{d}{H}}$, $W_O^l \in \mathbb{R}^{d \times d}$, $W_1^l \in \mathbb{R}^{d \times 4d}$ and $W_2^l \in \mathbb{R}^{4d \times d}$ are learnable projection weights. We hide the bias terms for simplification. $X^0$ ($X^l$ with $l = 0$) is normally set as $E^0$, and will be added with an extra learnable token $c^0$ when performing classification tasks, $i.e.$, $X^0 = [c^0; E^0]$. The feature is extracted from the same location of the last-layer sequence, which is $f = \mathrm{LN}(c^L)$.

Transformer has gained remarkable success due to its strong generalization capabilities, making it adaptable to a wide range of computer vision and natural language processing tasks (Devlin et al., 2019; Dosovitskiy et al., 2021). Notably, the recent vision-language pre-training model, CLIP, further underscores its efficacy by demonstrating impressive zero-shot performance (Radford et al., 2021). When processing an image $x$ and considering $K$ candidate classes, CLIP first generates the textual prompts for each of the $K$ classes. These prompts are descriptive phrases, such as "a photo of a cat" or "a photo of a dog". CLIP extracts corresponding textual features $f_{T_1}, \cdots, f_{T_K}$ and the image feature $f_I$. To predict the label for the given image, CLIP computes the cosine similarity between the image feature $f_I$ and each of the class prompts' features:

$$y_{\mathrm{pred}} = \arg\max_{k \in [K]} \frac{f_I^\top f_{T_k}}{\|f_I\|_2 \|f_{T_k}\|_2} \tag{4}$$

**Zero-shot CLIP for long-tailed recognition.** In Figures 2a to 2c, we evaluate the performance of zero-shot CLIP on long-tailed benchmark datasets. We discover that zero-shot CLIP outperforms most well-designed methods. Furthermore, by freezing the pre-trained backbone and introducing an additional classifier, the performance can be further improved, thereby highlighting the robustness of the representations learned by CLIP.

While zero-shot CLIP exhibits impressive performance on ImageNet-LT and Places-LT, its performance on another dataset, iNaturalist 2018, has been observed to be suboptimal. iNaturalist 2018 dataset poses a fine-grained long-tailed challenge, featuring a hierarchical categorization system spanning from 7 kingdoms to 8142 species. Although zero-shot CLIP underperforms in predicting the fine-grained species, it achieves high accuracy for predicting coarse-grained categories like "kingdom" and "phylum". Furthermore, by simply introducing an additional classifier, the accuracy increases from 4.2% to 57.9%. This underscores the utility of the representations extracted from CLIP when appropriately leveraged.

### 3.2 CLIP FINE-TUNING HURTS TAIL-CLASS PERFORMANCE

While the pre-trained model demonstrates commendable performance on downstream long-tailed recognition tasks, it does exhibit limitations. As shown in Figures 2d and 2e, zero-shot CLIP achieves high accuracy for tail classes, but falls short in achieving strong accuracy for head classes

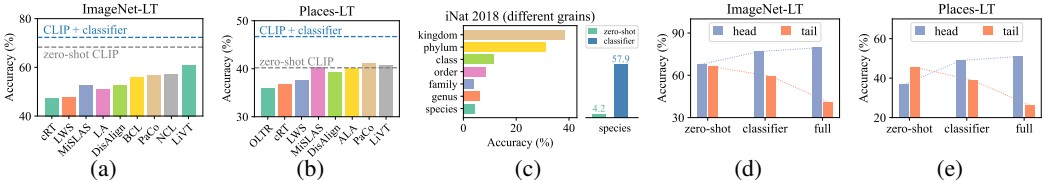

Figure 2: (a-b) On ImageNet-LT and Places-LT, zero-shot CLIP has surpassed many prior methods. (c) However, on iNaturalist 2018, zero-shot CLIP encounters challenges in achieving high accuracy for fine-grained categories. By simply introducing an additional classifier, the accuracy for species increases significantly. (d-e) Learning classifiers or full fine-tuning improves head-class accuracy while decreasing tail-class accuracy, even if we optimize the balanced LA loss.

because it does not effectively leverage downstream training datasets to improve the performance. A direct approach to address this problem is to fine-tune the CLIP model by keeping the backbone fixed and introducing a dedicated classifier. Figures 2d and 2e show that this approach yields notable improvements in head-class accuracy, at the expense of a reduction in tail-class accuracy. Another commonly adopted method is full fine-tuning, where the entire set of model parameters is updated. Nevertheless, full fine-tuning often comes with significant computational overheads on large-scale datasets. Moreover, it tends to encounter severe overfitting on long-tailed datasets due to the limited amount of tail-class samples. Note that we have already optimized Logit-Adjusted (LA) loss (Menon et al., 2021) for balanced prediction. In addition, we quantitatively assess the overfitting issue of CLIP fine-tuning methods. Due to space constraints, we present the results in Appendix A.

To mitigate the issue of performance deterioration on tail classes, recent approaches have explored two-stage training procedures (Ma et al., 2021) or the inclusion of additional training data (Tian et al., 2022). However, these strategies often introduce significant training overhead or require external training data, thereby limiting their practicality. In response to this, we introduce PEL, a simple yet effective parameter-efficient long-tailed fine-tuning method tailored for long-tailed recognition.

### 3.3 HOW PARAMETER-EFFICIENT FINE-TUNING HELPS?

**Mitigating overfitting via PEFT.**    In this paper, we explore parameter-efficient fine-tuning (PEFT) (Yu et al., 2023) which freezes the pre-trained model while introducing a few learnable parameters for the adaptation. This approach prevents the decline of generalization ability benefiting from image-text pre-training. As only a small set of task-specific parameters is introduced, the model not only mitigates overfitting but also exhibits rapid convergence. Concretely, we propose a unified framework capable of accommodating various PEFT methods. This framework is versatile and inclusive, allowing for the incorporation of a range of PEFT methods, including but not limited to

- *LN tuning* (Kim et al., 2021) adjusts the scaling and shifting parameters $\boldsymbol{\gamma}$ and $\boldsymbol{\beta}$ for LN modules, since an LN module can be formulated as $\text{LN}(\boldsymbol{X}) = \text{Normalize}(\boldsymbol{X}) \circ \boldsymbol{\gamma} + \boldsymbol{\beta}$.

- *Bias-terms Fine-tuning* (*BitFit*) (Zaken et al., 2022) aims to fine-tune only the bias parts of the model. Formally, given a projection function $\boldsymbol{X}\boldsymbol{W} + \boldsymbol{b}$, it freezes $\boldsymbol{W}$ and optimizes $\boldsymbol{b}$.

- *Visual Prompt Tuning* (*VPT*) (Jia et al., 2022) prepends learnable prompts $\boldsymbol{P}^l \in \mathbb{R}^{p \times d}$ at each layer to extend $\boldsymbol{X}^l = [\boldsymbol{c}^l; \boldsymbol{E}^l]$ to $[\boldsymbol{c}^l; \boldsymbol{P}^l; \boldsymbol{E}^l]$. It has two variations: 1) *VPT-Shallow*, which only prepends prompts at the first layer; 2) *VPT-Deep*, which prepends prompts at all layers.

- *Adapter* (Houlsby et al., 2019) proposes to optimize a bottleneck module. The definition is $\text{Adapter}(\boldsymbol{X}) = \text{ReLU}(\text{LN}(\boldsymbol{X})\boldsymbol{W}_{\text{down}})\boldsymbol{W}_{\text{up}}$, where $\boldsymbol{W}_{\text{down}} \in \mathbb{R}^{d \times r}$ and $\boldsymbol{W}_{\text{up}} \in \mathbb{R}^{r \times d}(r \ll d)$. In practical, it can be appended to the FFN layer to reconstruct $\text{FFN}(\cdot)$ to $\text{Adapter}(\text{FFN}(\cdot))$.

- *Low-Rank Adapter (LoRA)* (Hu et al., 2022) is applied to the MSA module. Specifically, it optimizes $\boldsymbol{W}_{\text{down}}$ and $\boldsymbol{W}_{\text{up}}$ to update $\boldsymbol{W}$ (*e.g.,* $\boldsymbol{W}_Q$ or $\boldsymbol{W}_V$) to $\boldsymbol{W} + \boldsymbol{W}_{\text{down}}\boldsymbol{W}_{\text{up}}$.

- *AdaptFormer* (Chen et al., 2022) changes the sequential *Adapter* to a parallel one. Formally, it computes $s \cdot \text{Adapter}(\hat{\boldsymbol{X}}^l)$ and adds it to $\boldsymbol{X}^l$ in Eq. (1). Here $s$ is a learnable scaling parameter.

Without loss of generality, we default to use *AdaptFormer* considering its state-of-the-art performance. We provide an overview of the proposed framework in Figure 3.

Apart from the PEFT module, an additional classifier is essential for adaptation. The linear classifier is widely employed due to its simplicity and versatility. Given a feature vector $\boldsymbol{f}$, the predicted logit

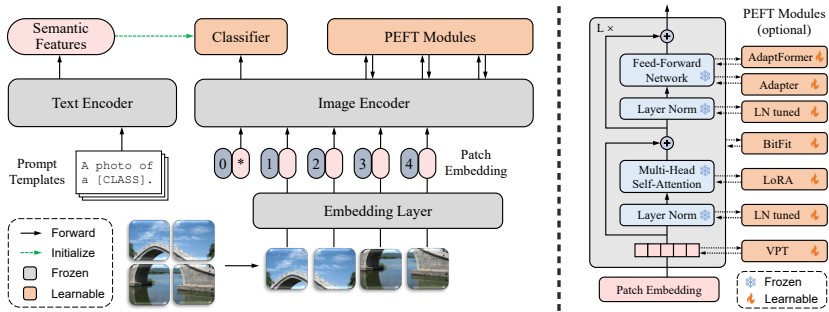

Figure 3: An overview of the framework PEL. Left: Reconstruct CLIP by employing PEFT and semantic-aware initialization. Right: Transformer-based image encoder with typical PEFT modules.

for class $k$ is computed as $z_k = \boldsymbol{w}_k^\top \boldsymbol{f} + b$. However, when training with long-tailed data, the norms of classifier weight $\boldsymbol{w}_k$ tend to exhibit imbalanced distribution, which can lead to biased predictions (Kang et al., 2020). To mitigate this issue, several approaches (Liu et al., 2020; Wu et al., 2021) propose to use the cosine classifier $z_k = \sigma \cdot \frac{\boldsymbol{w}_k^\top \boldsymbol{f}}{\|\boldsymbol{w}_k\|_2 \|\boldsymbol{f}\|_2}$. Here, $\sigma$ is a scaling factor, which is equivalent to dividing by a temperature $\tau$ in some literature. To underline the effectiveness of PEL, we opt for the logit-adjusted (LA) loss for training a cosine classifier:

$$\mathcal{L}_{\text{LA}}(\boldsymbol{x}, y = j) = -\log \frac{\exp(z_j + \log \mathrm{P}(y = j))}{\sum_{k \in [K]} \exp(z_k + \log \mathrm{P}(y = k))} \tag{5}$$

Here, $y = j$ represents the ground-truth label of $\boldsymbol{x}$ and $z_j$ is the predicted logit. $\mathrm{P}(y = k)$ signifies the class prior probability, which can be estimated based on the training data. We delve deeper into the theoretical understanding of the LA loss in Appendix B.

**Accelerating convergence via semantic-aware initialization.** When adapting the CLIP model using PEFT, it discards valuable knowledge embedded within the text modality, which inherently contains rich semantic information. While CLIP possesses strong visual feature extraction capabilities for downstream tasks, its classifier is not optimized for these tasks. To better capture potential relationships between classes and accelerate convergence, we propose to leverage the embedded semantic knowledge. Specifically, our approach involves using textual features associated with class labels to initialize the classifier weights. We generate hand-crafted textual prompts (*e.g.*, "a photo of a [CLASS].") and compute their features $\boldsymbol{f}_{T_1}, \cdots, \boldsymbol{f}_{T_K}$, which are then employed to initialize the classfier weights $\boldsymbol{w}_1, \cdots, \boldsymbol{w}_K$.

In contrast to previous CLIP fine-tuning methods like BALLAD (Ma et al., 2021) and VL-LTR (Tian et al., 2022), our approach stands out in its simplicity and efficiency. Unlike prior methods that use the text encoder in complex loss optimization processes, we solely rely on a single forward pass of the text encoder for each class description. This simple approach allows us to achieve a better initial state of the classifier without adding any computational overhead. Moreover, by harnessing the semantic knowledge inherent in class labels, our method leads to significant improvements in generalization. Unlike other initialization techniques such as linear probing and class mean features, our approach benefits from this semantic-aware information.

**Test-time ensembling can improve the generalization.** In the context of deep neural networks (DNNs), it is well-established that applying random perturbations to each input can lead to improved generalization. This principle is particularly crucial for Vision Transformers, where an image is divided into multiple patches, potentially resulting in the segmentation of continuous patterns into different patches. In this paper, we propose an enhancement to generalization by aggregating the predictions from a set of perturbed versions of the input image. Formally, given a test data point $\boldsymbol{x}$, its predicted logits $\boldsymbol{z}$ are obtained by averaging the predictions from $M$ perturbed versions:

$$\boldsymbol{z} = \log \mathrm{P}(\boldsymbol{y} \mid \boldsymbol{x}) = \frac{1}{M} \sum_{i=1}^{M} \log \mathrm{P}(\boldsymbol{y} \mid \alpha_i(\boldsymbol{x})) \tag{6}$$

Here, $\alpha_i(\boldsymbol{x})$ represents different perturbed versions of $\boldsymbol{x}$. In practice, we employ different image cropping positions as these perturbations (see Appendix C for further details). This approach helps mitigate bias introduced by image cropping. We term this technique *test-time ensembling* (TTE), and it can be seamlessly integrated into existing frameworks with minimal computational overhead.

## 4 EMPIRICAL STUDY

### 4.1 EXPERIMENTAL SETTINGS

**Datasets and evaluation.** We conduct experiments on four long-tailed datasets, including ImageNet-LT (Liu et al., 2019), Places-LT (Liu et al., 2019), iNaturalist 2018 (Van Horn et al., 2018) and CIFAR-100-LT (Cao et al., 2019). ImageNet-LT has 115.8K images from 1000 classes, with a maximum of 1280 and a minimum of 5 images per class. Places-LT contains 62.5K images from 365 classes, from a maximal 4980 to a minimum of 5 images per class. iNaturalist 2018 consists of 437.5K images distributed across 8142 species, with the number of images per species varying from as few as 2 to as many as 1000. In addition to measuring overall accuracy, we adhere to the evaluation protocol introduced by Liu et al. (2019) to report accuracy across three splits of classes: many-shot ($>$100 images), medium-shot (20$\sim$100 images), and few-shot ($<$20 images). Due to limited space, the results for CIFAR-100-LT are presented in Appendix A.

**Implementation details.** For all experiments, we use the SGD optimizer with a batch size of 128, weight decay of $5 \times 10^{-4}$, and momentum of 0.9. For parameter-efficient fine-tuning methods, the learning rate is 0.01. For full fine-tuning, we search the learning rate from $\{0.02, 0.01, 0.005, 0.002, 0.001, 0.0005\}$ considering its weak stability. For ImageNet-LT and Places-LT, we train the model for only 10 epochs; and for iNaturalist 2018, we train 20 epochs considering it has much more data. We set the bottleneck dimension $r = 2^{\lfloor \log_2 \left( \frac{K}{2L} \right) \rfloor}$ for AdaptFormer such that it learns even fewer parameters than the classifier (see Appendix E for detailed analysis). The scaling factor $\sigma$ of the cosine classifier is set to 25. All experiments are conducted on a single NVIDIA A800 GPU. In fact, a GPU with 20GB of memory is sufficient for all reproduction.

### 4.2 COMPARISON WITH STATE-OF-THE-ART METHODS

**Results on ImageNet-LT.** We report the test accuracy in Table 1. While existing approaches such as VL-LTR (Tian et al., 2022) and GML (Suh & Seo, 2023) rely on extensive auxiliary data to facilitate fine-tuning, our method PEL achieves superior performance by leveraging the test-time ensembling (TTE) technique alone. The use of external data not only incurs significant computational overhead but also reduces practicality due to the unavailability of such data in many real-world applications. The advantage of PEL is more significant compared with methods that do not use auxiliary data, *i.e.,*PEL surpasses the previous best method by 1.3% in accuracy. Importantly, PEL only needs 10 epochs of training and fine-tunes far fewer model parameters (*i.e.,* from 21.26M to 0.62M). It is worth noting that we do not include LPT (Dong et al., 2023) for comparison since it is pre-trained on ImageNet-21K, its results on ImageNet-LT were not reported in the original paper.

**Results on Places-LT.** From Table 2, we can see that PEL outperforms existing methods by a larger margin than that on ImageNet-LT. Even without TTE, PEL surpasses VL-LTR and RAC which use external training data by 1.4% in accuracy. By integrating TTE, the number increases to 2.1%. Similar to ImageNet-LT, we only need 10 epochs of training in contrast to 80 epochs (40 in each stage) for LPT. The amount of tunable parameters is also much fewer, *i.e.,* 0.18M. Nevertheless, PEL outperforms LPT by 1.4% in accuracy. When taking a closer look, we can see that PEL significantly improves the tail class performance, *i.e.,* from 46.9 to 50.5.

**Results on iNaturalist 2018.** We report the results in Table 3. Overall, our method achieves the best performance on this challenging dataset, surpassing LPT, VL-LTR, and RAC. We acknowledge that Decoder (Wang et al., 2023) uses fewer training epochs, however, its performance trails far behind PEL. Particularly, PEL (without using TTE) improves LPT by 3% in accuracy and PEL only needs 20 epochs of training compared with 160 epochs (80 per stage) for LPT. Although LPT uses fewer learnable parameters, we can reduce the parameters of PEL to reach a comparable quantity (*i.e.,* reduce the bottleneck dimension $r$ to 64, more details are given in Figure 6). In this case, PEL

Table 1: Comparison with state-of-the-art methods on ImageNet-LT.

| Methods | Backbone | Learnable Params. | #Epochs | Overall | Many | Medium | Few |
|---|---|---|---|---|---|---|---|
| **Training from scratch** | | | | | | | |
| cRT (Kang et al., 2020) | ResNet-50 | 23.51M | 90+10 | 47.3 | 58.8 | 44.0 | 26.1 |
| LWS (Kang et al., 2020) | ResNet-50 | 23.51M | 90+10 | 47.7 | 57.1 | 45.2 | 29.3 |
| MiSLAS (Zhong et al., 2021) | ResNet-50 | 23.51M | 180+10 | 52.7 | 62.9 | 50.7 | 34.3 |
| LA (Menon et al., 2021) | ResNet-50 | 23.51M | 90 | 51.1 | - | - | - |
| DisAlign (Zhang et al., 2021) | ResNet-50 | 23.51M | 90 | 52.9 | 61.3 | 52.2 | 31.4 |
| BCL (Zhu et al., 2022) | ResNet-50 | 23.51M | 100 | 56.0 | - | - | - |
| PaCo (Cui et al., 2021) | ResNet-50 | 23.51M | 400 | 57.0 | - | - | - |
| NCL (Li et al., 2022a) | ResNet-50 | 23.51M | 400 | 57.4 | - | - | - |
| LiVT (Xu et al., 2023) | ViT-B/16 | 85.80M | 100 | 60.9 | 73.6 | 56.4 | 41.0 |
| **Fine-tuning pre-trained model** | | | | | | | |
| BALLAD (Ma et al., 2021) | ViT-B/16 | 149.62M | 50+10 | 75.7 | 79.1 | 74.5 | 69.8 |
| Decoder (Wang et al., 2023) | ViT-B/16 | 21.26M | ∼18 | 73.2 | - | - | - |
| PEL (Ours) | ViT-B/16 | **0.62M** | **10** | 77.0 | 80.2 | 76.1 | 71.5 |
| PEL w/ TTE (Ours) | ViT-B/16 | **0.62M** | **10** | **78.3** | **81.3** | **77.4** | **73.4** |
| *Fine-tuning with Extra Data* | | | | | | | |
| *VL-LTR (Tian et al., 2022)* | *ViT-B/16* | *149.62M* | *100* | *77.2* | *84.5* | *74.6* | *59.3* |
| *GML (Suh & Seo, 2023)* | *ViT-B/16* | *149.62M* | *100* | *78.0* | *-* | *-* | *-* |

Table 2: Comparison with state-of-the-art methods on Places-LT.

| Methods | Backbone | Learnable Params. | #Epochs | Overall | Many | Medium | Few |
|---|---|---|---|---|---|---|---|
| **Training from scratch (with an ImageNet-1K pre-trained backbone)** | | | | | | | |
| OLTR (Liu et al., 2019) | ResNet-152 | 58.14M | 30 | 35.9 | 44.7 | 37.0 | 25.3 |
| cRT (Kang et al., 2020) | ResNet-152 | 58.14M | 90+10 | 36.7 | 42.0 | 37.6 | 24.9 |
| LWS (Kang et al., 2020) | ResNet-152 | 58.14M | 90+10 | 37.6 | 40.6 | 39.1 | 28.6 |
| MiSLAS (Zhong et al., 2021) | ResNet-152 | 58.14M | 90+10 | 40.4 | 39.6 | 43.3 | 36.1 |
| DisAlign (Zhang et al., 2021) | ResNet-152 | 58.14M | 30 | 39.3 | 40.4 | 42.4 | 30.1 |
| ALA (Zhao et al., 2022) | ResNet-152 | 58.14M | 30 | 40.1 | 43.9 | 40.1 | 32.9 |
| PaCo (Cui et al., 2021) | ResNet-152 | 58.14M | 30 | 41.2 | 36.1 | 47.9 | 35.3 |
| LiVT (Xu et al., 2023) | ViT-B/16 | 85.80M | 100 | 40.8 | 48.1 | 40.6 | 27.5 |
| **Fine-tuning pre-trained model** | | | | | | | |
| BALLAD (Ma et al., 2021) | ViT-B/16 | 149.62M | 50+10 | 49.5 | 49.3 | 50.2 | 48.4 |
| Decoder (Wang et al., 2023) | ViT-B/16 | 21.26M | ∼34 | 46.8 | - | - | - |
| LPT (Dong et al., 2023) | ViT-B/16 | 1.01M | 40+40 | 50.1 | 49.3 | 52.3 | 46.9 |
| PEL (Ours) | ViT-B/16 | **0.18M** | **10** | 51.5 | 51.3 | 52.2 | 50.5 |
| PEL w/ TTE (Ours) | ViT-B/16 | **0.18M** | **10** | **52.2** | **51.7** | **53.1** | **50.9** |
| *Fine-tuning with Extra Data* | | | | | | | |
| *VL-LTR (Tian et al., 2022)* | *ViT-B/16* | *149.62M* | *100* | *50.1* | *54.2* | *48.5* | *42.0* |
| *RAC (Long et al., 2022)* | *ViT-B/16* | *85.80M* | *30* | *47.2* | *48.7* | *48.3* | *41.8* |

achieves an accuracy of 77.7% (without TTE) / 79.0% (with TTE), which still outperforms LPT. In fact, due to the large number of classes of iNaturalist 2018, the classifier already contains 6.25M parameters. Therefore, the parameter quantity of PEL does not lead to too much cost.

## 4.3 COMPONENT ANALYSIS AND ABLATION STUDIES

**PEL improves representation separability.** Compared with the original CLIP, PEL embeds a few task-specific learnable parameters into each Transformer block. We show that this helps improve the representation separability among different classes. Figure 4 visualizes the cosine similarity between pairs of class mean features, the left one is produced by the original CLIP while the right one is produced by our method. From the plots, we can easily find that directly using CLIP representations

Table 3: Comparison with state-of-the-art methods on iNaturalist 2018.

| Methods | Backbone | Learnable Params. | #Epochs | Overall | Many | Medium | Few |
|---|---|---|---|---|---|---|---|
| **Training from scratch** | | | | | | | |
| cRT (Kang et al., 2020) | ResNet-50 | 23.51M | 90+10 | 65.2 | 69.0 | 66.0 | 63.2 |
| LWS (Kang et al., 2020) | ResNet-50 | 23.51M | 90+10 | 65.9 | 65.0 | 66.3 | 65.5 |
| MiSLAS (Zhong et al., 2021) | ResNet-50 | 23.51M | 200+30 | 71.6 | 73.2 | 72.4 | 70.4 |
| DiVE (He et al., 2021) | ResNet-50 | 23.51M | 90 | 69.1 | 70.6 | 70.0 | 67.6 |
| DisAlign (Zhang et al., 2021) | ResNet-50 | 23.51M | 90 | 69.5 | 61.6 | 70.8 | 69.9 |
| ALA (Zhao et al., 2022) | ResNet-50 | 23.51M | 90 | 70.7 | 71.3 | 70.8 | 70.4 |
| RIDE (Wang et al., 2021) | ResNet-50 | 23.51M | 100 | 72.6 | 70.9 | 72.4 | 73.1 |
| BCL (Zhu et al., 2022) | ResNet-50 | 23.51M | 100 | 71.8 | - | - | - |
| PaCo (Cui et al., 2021) | ResNet-50 | 23.51M | 400 | 73.2 | 70.4 | 72.8 | 73.6 |
| NCL (Li et al., 2022a) | ResNet-50 | 23.51M | 400 | 74.2 | 72.0 | 74.9 | 73.8 |
| GML (Suh & Seo, 2023) | ResNet-50 | 23.51M | 400 | 74.5 | - | - | - |
| LiVT (Xu et al., 2023) | ViT-B/16 | 85.80M | 100 | 76.1 | **78.9** | 76.5 | 74.8 |
| **Fine-tuning pre-trained model** | | | | | | | |
| Decoder (Wang et al., 2023) | ViT-B/16 | 21.26M | ∼5 | 59.2 | - | - | - |
| LPT (Dong et al., 2023) | ViT-B/16 | **1.01M** | 80+80 | 76.1 | - | - | 79.3 |
| PEL (Ours) | ViT-B/16 | 4.75M | 20 | 79.1 | 72.4 | 79.0 | 81.1 |
| PEL w/ TTE (Ours) | ViT-B/16 | 4.75M | 20 | **80.4** | 74.0 | **80.3** | **82.2** |
| **Fine-tuning with Extra Data** | | | | | | | |
| VL-LTR (Tian et al., 2022) | ViT-B/16 | 149.62M | 100 | 76.8 | - | - | - |
| RAC (Long et al., 2022) | ViT-B/16 | 85.80M | 20 | 80.2 | 75.9 | 80.5 | 81.1 |

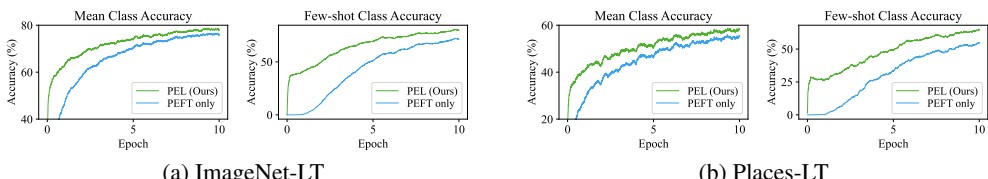

(a) ImageNet-LT     (b) Places-LT     (c) iNaturalist 2018

Figure 4: Visualization of the cosine similarities of class mean features on three datasets. For each dataset, the left is produced using the original CLIP, and the right is by our method.

(a) ImageNet-LT          (b) Places-LT

Figure 5: Convergence curve of mean class and few-shot class training accuracy.

does not provide much discriminative information for the classifier. In contrast, our method recovers the diagonal, showing that it can learn useful features that separate distinct classes. Due to the large amount of classes in iNaturalist 2018, the diagonal is more obvious if we zoom in on the plot.

**PEL improves convergence.** Figure 5 presents the mean class and few-shot class training accuracy as a function of epochs. Overall, we can observe that PEL converges rapidly with 10 training epochs on both datasets. As expected, the semantic-aware initialization attributes to the fast convergence, especially in the case of the tail classes.

**On parameter-efficient fine-tuning methods.** PEL is a general framework in which many existing parameter-efficient fine-tuning (PEFT) methods can be integrated. In addition to commonly used full fine-tuning and classifier fine-tuning, we test PEL with other 7 types of PEFT methods in Table 4. Overall, the integration of most PEFT methods yields enhanced performance. Specifically, AdaptFormer performs best on both datasets and Adapter achieves slightly low accuracy (*i.e.,* 0.2%).

Table 4: Comparison of different fine-tuning methods. All methods use semantic-aware initialization for the classifier (if have) and test-time ensemble for fair comparison.

| Methods | | ImageNet-LT | | | | Places-LT | | | |
| --- | --- | --- | --- | --- | --- | --- | --- | --- | --- |
| | | Overall | Many | Medium | Few | Overall | Many | Medium | Few |
| Zero-shot CLIP | | 68.3 | 69.2 | 67.6 | 67.7 | 40.2 | 38.3 | 39.2 | 45.9 |
| Full fine-tuning | | 74.4 | **82.2** | 73.9 | 53.8 | 47.2 | 51.6 | 48.5 | 36.2 |
| Classifier fine-tuning | | 74.0 | 77.9 | 73.9 | 62.8 | 49.3 | 50.0 | 50.9 | 44.2 |
| PEL w/ | LN tuning | 73.5 | 76.9 | 72.5 | 67.1 | 50.5 | 50.2 | 51.6 | 48.3 |
| | BitFit | 77.0 | 79.7 | 76.3 | 71.9 | 51.5 | 51.2 | 52.3 | 50.0 |
| | VPT-shallow | 75.2 | 78.8 | 74.8 | 66.8 | 49.9 | 50.5 | 51.4 | 45.3 |
| | VPT-deep | 77.2 | 79.5 | 76.5 | 72.8 | 51.5 | 51.4 | 52.3 | 49.8 |
| | Adapter | 78.1 | 81.3 | 77.1 | 72.8 | 52.0 | **51.7** | 52.7 | 51.0 |
| | LoRA | 76.9 | 79.6 | 76.2 | 71.7 | 51.8 | 51.5 | 52.5 | 50.5 |
| | AdaptFormer | **78.3** | 81.3 | **77.4** | **73.4** | **52.2** | **51.7** | **53.1** | **50.9** |

Table 5: Comparison of PEL with different classifier initialization methods. All methods use the test-time ensemble for fair comparisons.

| Methods | ImageNet-LT | | | | Places-LT | | | |
| --- | --- | --- | --- | --- | --- | --- | --- | --- |
| | Overall | Many | Medium | Few | Overall | Many | Medium | Few |
| Random initialization | 76.1 | 80.8 | 75.9 | 63.2 | 49.6 | 51.2 | 52.1 | 41.0 |
| Linear probing | 77.1 | **81.8** | 76.4 | 66.6 | 49.9 | 51.2 | 51.0 | 45.0 |
| Class mean features | 77.5 | 81.3 | 76.8 | 69.4 | 51.3 | 51.6 | 52.1 | 48.8 |
| Semantic-aware initialization | **78.3** | 81.3 | **77.4** | **73.4** | **52.2** | **51.7** | **53.1** | **50.9** |

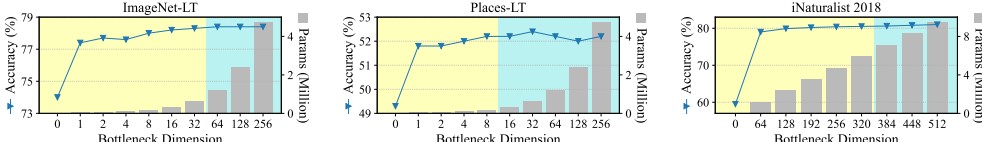

Figure 6: Comparison of different learnable parameters by changing the bottleneck dimension $r$. In the yellow area, the PEFT module has fewer learnable parameters than the classifier. While in the blue area, the quantity of parameters of the PEFT module surpasses the classifier.

**Effect of semantic-aware initialization.** In Table 5, we test three kinds of classifier initialization strategies in comparison with the random initialization baseline. In PEL, we use the textual feature by default because it transfers semantic relations between classes during fine-tuning. Another intuitive strategy using the class mean features to initialize the classifier achieves slightly poor performance but still significantly improves the baseline. This experiment shows that a good starting point for parameter optimization can lead to a better solution and faster convergence.

**Impact of the quantity of learnable parameters.** In PEL, we can define the amounts of learnable parameters. In Figure 6, we study how much the parameters impact performance by controlling the bottleneck dimension $r$. Overall, we find that the performance is robust to the change of dimensions and it achieves the best results when employing comparable learnable parameters to the classifier.

## 5 CONCLUSION

In summary, this paper introduces PEL, a straightforward yet effective framework for fine-tuning the CLIP model on long-tailed datasets. The proposed framework is versatile, allowing for the integration of various parameter-efficient fine-tuning methods, and notably achieves convergence in fewer than 20 training epochs. Importantly, our method does not rely on the availability of external training datasets, setting it apart from previous approaches. Despite its simplicity, our method consistently outperforms numerous baselines across a range of datasets, including CIFAR-100-LT, ImageNet-LT, Places-LT, and iNaturalist 2018. We emphasize the ease of training and hope that our approach serves as an inspiration for further advancements in the field of long-tailed recognition.

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

# A  ADDITIONAL EXPERIMENTS

Due to the page limitation, we report additional experimental results here, including 1) results on CIFAR-100-LT dataset; 2) additional results on iNaturalist 2018; 3) Partial fine-tuning vs. parameter-efficient fine-tuning; 4) effects of the AdaptFormer module on different layers; 5) quantifying the overfitting issue of fine-tuning methods; 6) more detailed observations on model convergence; 7) comparison of different classifiers; 8) comparison of different losses; 9) ablation study of components; 10) PEL with ResNet as backbone.

**Results on CIFAR-100-LT dataset.**    CIFAR-100-LT (Cao et al., 2019) is derived from CIFAR-100 (Krizhevsky et al., 2009) and is constructed with various imbalance ratios, including 100, 50, and 10. The implementation settings for CIFAR-100-LT remain consistent with those used for ImageNet-LT and Places-LT, as described in Section 4.1. Table 6 presents the results on CIFAR-100-LT. Following the experimental setup used for large-scale datasets, we fine-tune the CLIP model for 10 epochs. The results clearly demonstrate that PEL outperforms other methods, including LiVT, BALLAD, and various training-from-scratch approaches. These results hold true regardless of whether Test-Time Ensembling (TTE) is applied or not. Additionally, we extend our experiments by replacing the CLIP with ViT which is pre-trained on the ImageNet-21K dataset. Notably, ViT lacks a corresponding text encoder. In this case, we employ the class mean features to initialize the classifier. Despite the inherent class overlaps between ImageNet-21K and CIFAR-100, which naturally lead to higher performance, our method surpasses LPT with fewer training epochs and learnable parameters.

Table 6: Comparison with state-of-the-art methods on CIFAR-100-LT with various imbalance ratios. [†]Pre-trained model from ImageNet-21K[1] has several classes related to CIFAR-100[2], which potentially leads to data leakage.

| Methods | Backbone | Learnable Params. | #Epochs | Imbalance Ratio | | |
|---|---|---|---|---|---|---|
| | | | | 100 | 50 | 10 |
| **Training from scratch** | | | | | | |
| LDAM (Cao et al., 2019) | ResNet-32 | 0.46M | 200 | 42.0 | 46.6 | 58.7 |
| BBN (Zhou et al., 2020) | ResNet-32 | 0.46M | 200 | 42.6 | 47.0 | 59.1 |
| DiVE (He et al., 2021) | ResNet-32 | 0.46M | 200 | 45.4 | 51.1 | 62.0 |
| MiSLAS (Zhong et al., 2021) | ResNet-32 | 0.46M | 200+10 | 47.0 | 52.3 | 63.2 |
| BS (Ren et al., 2020) | ResNet-32 | 0.46M | 400 | 50.8 | 54.2 | 63.0 |
| PaCo (Cui et al., 2021) | ResNet-32 | 0.46M | 400 | 52.0 | 56.0 | 64.2 |
| BCL (Zhu et al., 2022) | ResNet-32 | 0.46M | 200 | 51.9 | 56.6 | 64.9 |
| **Fine-tuning pre-trained model** | | | | | | |
| LiVT (Xu et al., 2023) | ViT-B/16 | 85.80M | 100 | 58.2 | - | 69.2 |
| BALLAD (Ma et al., 2021) | ViT-B/16 | 149.62M | 50+10 | 77.8 | - | - |
| PEL (Ours) | ViT-B/16 | **0.10M** | **10** | 80.3 | 82.0 | 83.8 |
| PEL w/ TTE (Ours) | ViT-B/16 | **0.10M** | **10** | **81.7** | **83.1** | **84.9** |
| **Fine-tuning pre-trained model from ImageNet-21K[†]** | | | | | | |
| LPT (Dong et al., 2023) | ViT-B/16 | 1.01M | 40+40 | **89.1** | 90.0 | 91.0 |
| PEL (Ours) | ViT-B/16 | **0.10M** | **10** | **89.1** | **90.2** | **91.3** |

**Additional results on iNaturalist 2018.**    In PEL, we train 20 epochs on iNaturalist 2018 considering its large data scale. In Table 7, we report the results of training different epochs. When training for 5 epochs, PEL achieves an overall accuracy of 67.3% (w/o TTE) / 68.6% (w/ TTE), which surpasses Decoder (Wang et al., 2023) by more than 8% (please refer to Table 3 for comparison). Moreover, by training more epochs (*e.g.,* more than 30 epochs), PEL achieves an additional performance improvement by 1%. However, this will increase the computational overhead, so we abort this costly training approach in PEL.

---

[1]https://storage.googleapis.com/bit_models/imagenet21k_wordnet_lemmas.txt
[2]https://www.cs.toronto.edu/%7Ekriz/cifar.html

Table 7: Results of PEL (with and without TTE) on iNaturalist 2018 by training different epochs.

| Methods | #Epochs | Overall | Many | Medium | Few |
|---|---|---|---|---|---|
| PEL (Ours) | 5 | 67.3 | 70.4 | 71.0 | 61.8 |
| | 10 | 76.1 | 71.3 | 75.9 | 77.5 |
| | 20 | 79.1 | 72.4 | 79.0 | 81.1 |
| | 30 | 80.1 | 73.8 | 80.0 | 81.9 |
| | 40 | 80.2 | 74.4 | 79.9 | 82.1 |
| | 50 | 80.3 | 75.3 | 79.9 | 82.1 |
| PEL w/ TTE (Ours) | 5 | 68.6 | 70.5 | 72.3 | 63.5 |
| | 10 | 77.3 | 71.9 | 77.1 | 78.9 |
| | 20 | 80.4 | 74.0 | 80.3 | 82.2 |
| | 30 | 81.3 | 75.1 | **81.2** | 83.0 |
| | 40 | 81.3 | 75.0 | 81.1 | 83.3 |
| | 50 | **81.4** | **75.9** | 80.9 | **83.5** |

**Partial fine-tuning vs. parameter-efficient fine-tuning.** Partial fine-tuning (He et al., 2022) is an intuitive way to reduce the learnable parameters and avoid overfitting. Specifically, it fine-tunes the last $k$ layers of Transformer blocks while freezing the others. In Figure 7, we compare partial fine-tuning and PEL on ImageNet-LT, Places-LT, and iNaturalist 2018. Similar to full fine-tuning, partial fine-tuning is also sensitive to learning rate. When $k$ is small (*e.g.,* $0, 1, 2$), a higher learning rate is better. However, when $k$ is large (*e.g.,* $9, 12$), the high learning rate leads to a severe deterioration in the accuracy, thus a smaller learning rate is more appropriate. Moreover, even if we have searched for the optimal learning rate, it is non-trivial to choose the number of fine-tuned layer $k$ for different datasets, as the best $k$ is 2 for ImageNet-LT, 1 for Places-LT, and 6 for iNaturalist 2018. In contrast, PEL consistently performs well with fixed hyperparameters.

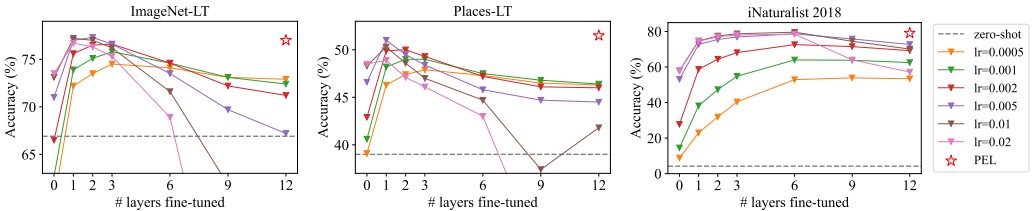

Figure 7: Partial fine-tuning the last $k$ layers. Both methods use cosine classifier and semantic-aware initialization for fair comparison. Similar to full fine-tuning, we search learning rate from $\{0.02, 0.01, 0.005, 0.002, 0.001, 0.0005\}$ for partial fine-tuning. For PEL, we fix the learning rate to 0.01. Partial fine-tuning needs to elaborately choose the proper learning rate and the fine-tuned layers for the best performance, while PEL consistently performs optimally.

**Effects of the AdaptFormer module on different layers.** In each layer, the output of the Adapt-Former module is multiplied by a learnable scaling parameter $s$ before being added to the corresponding block. Therefore, we can compare the values of $s$ to analyze the effects of AdaptFormer for different layers. In Figure 8, we visualize the learned $s$ from each layer. It is inspiring that Adapt-Former can adaptively learn suitable scaling parameters for different layers. Moreover, the values of the last layers tend to be larger, which indicates that the adaptation of the last several layers is more significant for downstream classification tasks.

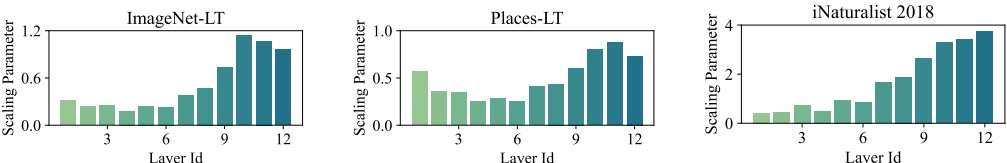

Figure 8: Learned scaling parameters of the PEFT modules (AdaptFormer) in different layers. AdaptFormer adaptively learns suitable scaling parameters for each layer, and the last several layers tend to have larger scaling parameters.

**Quantifying the overfitting issue of fine-tuning methods.** To give a more thorough comparison between classifier fine-tuning, full fine-tuning and PEL, we compute their training and test accuracy, as well as the accuracy gap, and report the results in Table 8. We highlight the overfitting results with red colors. The results show that full fine-tuning tends to cause overfitting, as the gaps between training and test accuracy are higher. Besides, classifier fine-tuning also cause overfitting regarding the tail-class performance.

Table 8: Training and test accuracy of different fine-tuning methods on ImageNet-LT, Places-LT, and iNaturalist 2018. The red number denotes that the gap between training and test accuracy is larger than PEL, which indicates overfitting.

(a) ImageNet-LT

| Methods | Overall | | | Many-shot | | | Medium-shot | | | Few-shot | | |
|---|---|---|---|---|---|---|---|---|---|---|---|---|
| | train | test | Δ | train | test | Δ | train | test | Δ | train | test | Δ |
| Classifier (*best lr*) | 83.4 | 73.5 | 9.9 | 83.2 | 76.6 | 6.6 | 86.9 | 72.8 | 14.1 | 91.7 | 67.2 | 24.5 |
| Classifier (*lr equal to PEL*) | 82.4 | 73.1 | 9.3 | 82.6 | 77.0 | 5.6 | 83.8 | 73.1 | 10.7 | 79.3 | 61.6 | 17.7 |
| Full (*best lr*) | 91.6 | 72.9 | 18.7 | 92.3 | 80.8 | 11.5 | 88.5 | 72.4 | 16.1 | 72.8 | 52.1 | 20.7 |
| Full (*lr equal to PEL*) | 91.6 | 61.7 | 29.9 | 91.8 | 70.3 | 21.5 | 91.3 | 60.0 | 21.3 | 86.0 | 43.4 | 42.6 |
| PEL (Ours) | 87.1 | 77.0 | 10.1 | 86.8 | 80.2 | 6.6 | 87.9 | 76.1 | 11.8 | 88.9 | 71.5 | 17.4 |

(b) Places-LT

| Methods | Overall | | | Many-shot | | | Medium-shot | | | Few-shot | | |
|---|---|---|---|---|---|---|---|---|---|---|---|---|
| | train | test | Δ | train | test | Δ | train | test | Δ | train | test | Δ |
| Classifier (*best lr*) | 56.8 | 48.5 | 8.3 | 55.0 | 48.6 | 6.4 | 66.3 | 49.1 | 17.2 | 79.1 | 46.8 | 32.3 |
| Classifier (*lr equal to PEL*) | 56.2 | 48.3 | 7.9 | 54.5 | 49.3 | 5.2 | 62.4 | 49.9 | 12.5 | 62.6 | 42.5 | 20.1 |
| Full (*best lr*) | 75.2 | 46.4 | 28.8 | 74.8 | 51.0 | 23.8 | 78.1 | 47.6 | 30.5 | 66.8 | 35.3 | 31.5 |
| Full (*lr equal to PEL*) | 75.7 | 41.8 | 33.9 | 77.0 | 46.3 | 30.7 | 85.7 | 43.1 | 42.6 | 82.0 | 30.5 | 51.5 |
| PEL (Ours) | 59.3 | 51.5 | 7.8 | 58.3 | 51.3 | 7.0 | 65.9 | 52.2 | 13.7 | 71.1 | 50.5 | 20.6 |

(c) iNaturalist 2018

| Methods | Overall | | | Many-shot | | | Medium-shot | | | Few-shot | | |
|---|---|---|---|---|---|---|---|---|---|---|---|---|
| | train | test | Δ | train | test | Δ | train | test | Δ | train | test | Δ |
| Classifier (*best lr*) | 60.2 | 57.9 | 2.3 | 53.7 | 46.7 | 7.0 | 74.7 | 56.8 | 17.9 | 85.4 | 62.1 | 23.3 |
| Classifier (*lr equal to PEL*) | 60.2 | 57.9 | 2.3 | 53.7 | 46.7 | 7.0 | 74.7 | 56.8 | 17.9 | 85.4 | 62.1 | 23.3 |
| Full (*best lr*) | 92.6 | 72.7 | 19.9 | 91.8 | 70.3 | 21.5 | 96.1 | 73.1 | 23.0 | 96.8 | 72.7 | 24.1 |
| Full (*lr equal to PEL*) | 87.8 | 70.1 | 17.7 | 85.1 | 63.7 | 21.4 | 95.8 | 70.0 | 25.8 | 98.7 | 71.8 | 26.9 |
| PEL (Ours) | 90.4 | 79.1 | 11.3 | 88.0 | 72.4 | 15.6 | 97.4 | 79.0 | 18.4 | 99.4 | 81.1 | 18.3 |

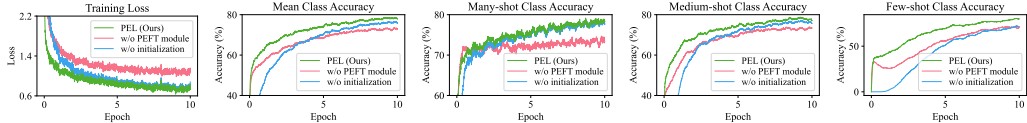

Figure 9: Convergence curves of training loss and accuracy on ImageNet-LT.

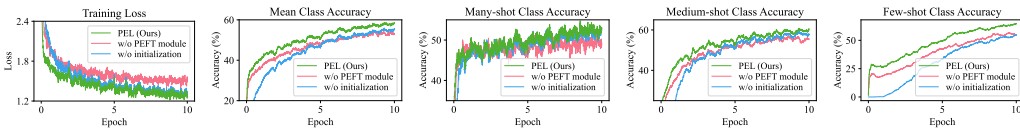

Figure 10: Convergence curves of training loss and accuracy on Places-LT.

**More detailed observations on model convergence.** In Figures 9 and 10, we illustrate the convergence curve on training loss and training accuracy. We report the mean accuracy of all classes, as well as many-shot, medium-shot, and few-shot classes. The results show that PEL converges rapidly with 10 training epochs on both head and tail classes. Without the PEFT module, the training loss and accuracy converge suboptimally on all classes. Without semantic-aware initialization, the head-class accuracy is slightly affected, while the tail-class accuracy decreases by a large margin.

**Comparison of different classifiers.** In PEL, we default to optimize the cosine classifier with a scaling factor $\sigma = 25$. One may be concerned about other classifiers or scaling factor $\sigma$. We further adopt the linear classifier $z_k = \boldsymbol{w}_k^\top \boldsymbol{f} + b$, L2-normalized classifier $z_k = \frac{\boldsymbol{w}_k^\top}{\|\boldsymbol{w}_k\|_2} \boldsymbol{f}$, as well as cosine classifier with $\sigma \in \{15, 20, 25, 30, 35\}$, and report the results in Table 9. The results show that the linear classifier performs well on ImageNet-LT and Places-LT, but unsatisfactorily on iNaturalist 2018. This can be inferred from the classifier weight norms shown in Figure 11, where the weight norms of iNaturalist 2018 are much more skewed. By removing the impact of weight norms, the L2-normalized classifier achieves higher performance, especially on the tail classes. When adopting the cosine classifier, setting $\sigma$ to 25 or 30 leads to the best performance. Without loss of generality, we default to set $\sigma = 25$.

Table 9: Performance of PEL with different classifiers. All methods use semantic-aware initialization and test-time ensemble for fair comparison.

| Classifiers | | ImageNet-LT | | | | Places-LT | | | | iNaturalist 2018 | | | |
|---|---|---|---|---|---|---|---|---|---|---|---|---|---|
| | | Overall | Many | Med. | Few | Overall | Many | Med. | Few | Overall | Many | Med. | Few |
| Linear | | 78.2 | 81.2 | 77.2 | 72.8 | **52.3** | 51.7 | 52.8 | 52.0 | 75.7 | **75.8** | 77.4 | 73.6 |
| L2-normalized | | 78.4 | 81.2 | 77.2 | **74.7** | 52.2 | 51.3 | 52.7 | **52.4** | 80.0 | 74.1 | 79.9 | 81.8 |
| Cosine | $\sigma = 15$ | 75.3 | 81.1 | 76.1 | 55.9 | 49.4 | **52.8** | 52.8 | 35.3 | 76.5 | 73.4 | 76.4 | 77.4 |
| | $\sigma = 20$ | 77.5 | 81.1 | 77.1 | 69.1 | 51.6 | 52.2 | **53.3** | 46.8 | 79.6 | 73.9 | 79.3 | 81.6 |
| | $\sigma = 25$ | 78.3 | 81.3 | **77.4** | 73.4 | 52.2 | 51.7 | 53.1 | 50.9 | **80.4** | 74.0 | 80.3 | **82.2** |
| | $\sigma = 30$ | **78.5** | **81.5** | 77.3 | 74.2 | 52.1 | 51.5 | 52.6 | 51.8 | 80.3 | 73.8 | **80.4** | 81.9 |
| | $\sigma = 35$ | 78.4 | **81.5** | 77.1 | 73.9 | 51.7 | 51.3 | 52.1 | 51.7 | 79.8 | 74.1 | 79.9 | 81.3 |

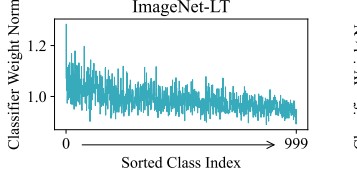
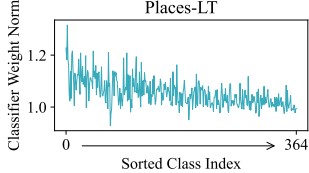
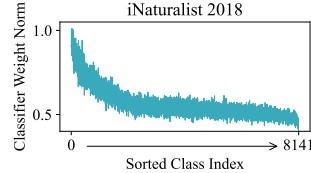

Figure 11: Weight norms of the learned linear classifier on three long-tailed datasets. Classes are sorted by their frequency in the training dataset. On iNaturalist 2018, the weight norms are much more imbalanced, leading to a suboptimal performance of the linear classifier.

Table 10: Performance of PEL with different losses. All methods use semantic-aware initialization and test-time ensemble for fair comparison.

| Losses | ImageNet-LT | | | | Places-LT | | | |
|---|---|---|---|---|---|---|---|---|
| | Overall | Many | Medium | Few | Overall | Many | Medium | Few |
| CE | 71.8 | 86.1 | 68.7 | 42.1 | 42.1 | 56.7 | 38.0 | 24.5 |
| Focal (Lin et al., 2017) | 72.1 | 85.5 | 69.1 | 44.7 | 42.7 | 56.1 | 38.8 | 26.8 |
| LDAM (Cao et al., 2019) | 69.6 | **86.4** | 66.5 | 33.4 | 40.4 | **56.8** | 36.0 | 20.1 |
| CB (Cui et al., 2019) | 76.9 | 82.3 | 76.3 | 63.5 | 50.0 | 52.6 | 51.5 | 41.9 |
| GRW (Zhang et al., 2021) | 76.9 | 82.3 | 76.3 | 63.7 | 50.1 | 52.4 | 51.7 | 42.0 |
| LADE (Hong et al., 2021) | 78.0 | 81.2 | 76.7 | **73.4** | 51.2 | 51.3 | 51.7 | 49.6 |
| LA (Menon et al., 2021) | **78.3** | 81.3 | **77.4** | **73.4** | **52.2** | 51.7 | **53.1** | **50.9** |

**Comparison of different losses.** In Table 10, we compare the performance of PEL with different loss functions, including cross-entropy (CE) loss, focal loss (Lin et al., 2017) proposed for hard

example mining, label-distribution-aware margin (LDAM) loss (Cao et al., 2019), class-balanced (CB) loss (Cui et al., 2019), generalized re-weighting (GRW) loss (Zhang et al., 2021), label distribution disentangling (LADE) loss (Hong et al., 2021). The results are shown in Table 10, which demonstrate that logit-adjusted (LA) loss achieves the highest performance. Furthermore, we give a theoretical analysis of logit-adjusted loss in Appendix B.

**Ablation study of components.** To assess the impact of each component, we conduct a systematical ablation study on different components in PEL including 1) the PEFT module, 2) the Logit-Adjusted (LA) loss, 3) Semantic-Aware Initialization (abbreviated as SAI), and 4) Test-Time Ensembling (TTE). The results presented in Table 11 demonstrate the effectiveness of each component. Specifically, 1) PEFT enhances performance on both head and tail classes; 2) without the LA loss, predictions tend to be biased to head classes; 3) Semantic-Aware Initialization (SAI) consistently improves the generalization, particularly on tail classes; 4) Test-Time Ensembling (TTE) further boosts the performance across both head and tail classes.

Table 11: Ablation study of each key component in PEL. The baseline involves learning a cosine classifier using Cross-Entropy (CE) loss.

| PEFT | LA | SAI | TTE | ImageNet-LT | | | | Places-LT | | | |
|------|-----|-----|-----|---------|------|--------|------|---------|------|--------|------|
| | | | | Overall | Many | Medium | Few | Overall | Many | Medium | Few |
| | | | | 60.9 | 82.6 | 56.7 | 13.8 | 34.3 | 53.6 | 30.5 | 7.5 |
| ✓ | | | | 68.9 | **84.7** | 66.3 | 33.7 | 38.9 | **55.6** | 35.3 | 16.4 |
| ✓ | ✓ | | | 74.9 | 79.7 | 74.7 | 61.7 | 48.7 | 50.6 | 50.8 | 40.5 |
| ✓ | ✓ | ✓ | | 77.0 | 80.2 | 76.1 | 71.5 | 51.5 | 51.3 | 52.2 | 50.5 |
| ✓ | ✓ | ✓ | ✓ | **78.3** | 81.3 | **77.4** | **73.4** | **52.2** | 51.7 | **53.1** | 50.9 |

**PEL with ResNet as the backbone** PEL incorporates Transformer-based models as its backbone. One may have concerns regarding adopting the widely used ResNet (He et al., 2016). However, since there is no off-the-shelf PEFT method specifically designed for ResNet, it is challenging to integrate our method with ResNet. Despite this constraint, we have explored some straightforward strategies, including 1) incorporating a scaling and shifting (SSF) (Lian et al., 2022) module after the backbone, and 2) fine-tuning solely the bias terms of ResNet. The results are presented below. In comparison to zero-shot CLIP and previous methods (reported in Tables 1 and 2), our method achieves significantly superior performance with lower computational costs.

Table 12: Results on ImageNet-LT with ResNet as the backbone. All methods use semantic-aware initialization and test-time ensemble for fair comparison.

| Methods | Backbone | Learnable Params. | #Epochs | Overall | Many | Medium | Few |
|---------|----------|-------------------|---------|---------|------|--------|------|
| Zero-shot CLIP | ResNet-50 | - | - | 57.6 | 58.6 | 56.9 | 56.9 |
| PEL w/ SSF | ResNet-50 | 0.002M | 10 | 66.9 | 72.0 | 66.5 | 54.1 |
| PEL w/ bias tuning | ResNet-50 | 0.034M | 10 | 67.8 | 72.0 | 67.3 | 57.6 |
| PEL w/ bias tuning & SSF | ResNet-50 | 0.036M | 10 | **68.3** | **72.5** | **67.8** | **58.2** |

Table 13: Results on Places-LT with ResNet as backbone. All methods use semantic-aware initialization and test-time ensemble for fair comparison.

| Methods | Backbone | Learnable Params. | #Epochs | Overall | Many | Medium | Few |
|---------|----------|-------------------|---------|---------|------|--------|------|
| Zero-shot CLIP | ResNet-50 | - | - | 35.2 | 33.1 | 34.6 | 40.4 |
| PEL w/ SSF | ResNet-50 | 0.002M | 10 | 46.7 | 47.5 | 48.7 | 40.7 |
| PEL w/ bias tuning | ResNet-50 | 0.034M | 10 | 47.9 | **48.2** | 49.8 | 42.9 |
| PEL w/ bias tuning & SSF | ResNet-50 | 0.036M | 10 | **48.1** | 48.1 | **50.0** | **43.9** |

**On representation separability of different PEFT methods** In Figures 12 and 13, we illustrate the impacts of different PEFT methods on representation separability. Compared to the origin CLIP, all of these PEFT methods contribute to more distinctive features. Notably, Adapter and Adapt-Former are more effective in enhancing separability. Conversely, LN tuning and VPT-shallow may yield relatively weaker effects, which is aligned with their inferior performance in Table 4.

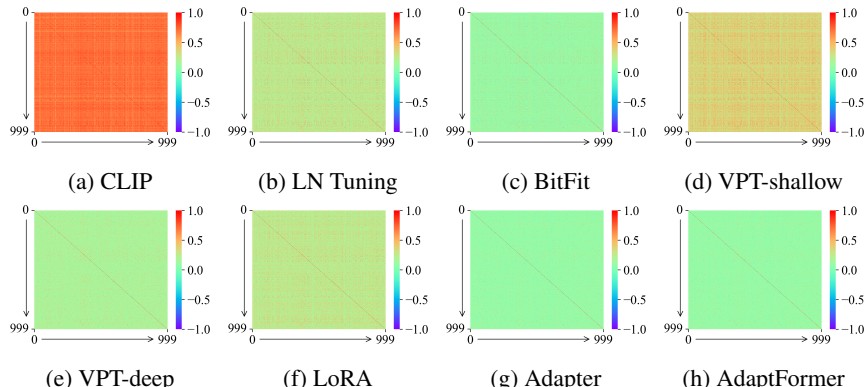

Figure 12: Visualization of the cosine similarities of class mean features on ImageNet-LT.

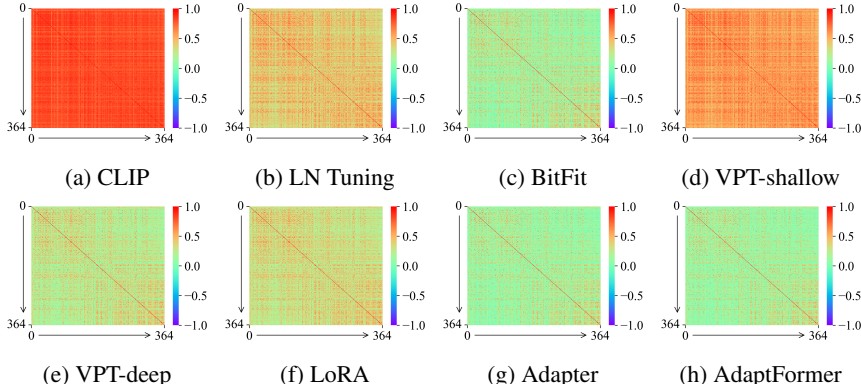

Figure 13: Visualization of the cosine similarities of class mean features on Places-LT.

# B UNDERSTANDING LOGIT-ADJUSTED LOSS

Logit-adjusted loss (Menon et al., 2021) (or termed Balanced-Softmax (Ren et al., 2020)) is widely used in previous literature (Hong et al., 2021; Zhao et al., 2022; Li et al., 2022b) because of its theoretical optimality and formal simplicity. Following we give a brief proof:

$$
\begin{aligned}
\mathrm{P}_s(y = j \mid \boldsymbol{x}) &= \frac{\mathrm{P}_s(y = j \mid \boldsymbol{x})}{\sum_{k \in [K]} \mathrm{P}_s(y = k \mid \boldsymbol{x})} \\
&= \frac{\mathrm{P}_t(y = j \mid \boldsymbol{x}) \cdot \dfrac{\mathrm{P}_s(y = j \mid \boldsymbol{x})}{\mathrm{P}_t(y = j \mid \boldsymbol{x})}}{\sum_{k \in [K]} \mathrm{P}_t(y = k \mid \boldsymbol{x}) \cdot \dfrac{\mathrm{P}_s(y = k \mid \boldsymbol{x})}{\mathrm{P}_t(y = k \mid \boldsymbol{x})}} \\
&= \frac{\mathrm{P}_t(y = j \mid \boldsymbol{x}) \cdot \dfrac{\mathrm{P}_s(\boldsymbol{x} \mid y = j)}{\mathrm{P}_t(\boldsymbol{x} \mid y = j)} \cdot \dfrac{\mathrm{P}_s(y = j)}{\mathrm{P}_t(y = j)} \cdot \dfrac{\mathrm{P}_t(\boldsymbol{x})}{\mathrm{P}_s(\boldsymbol{x})}}{\sum_{k \in [K]} \mathrm{P}_t(y = k \mid \boldsymbol{x}) \cdot \dfrac{\mathrm{P}_s(\boldsymbol{x} \mid y = k)}{\mathrm{P}_t(\boldsymbol{x} \mid y = k)} \cdot \dfrac{\mathrm{P}_s(y = k)}{\mathrm{P}_t(y = k)} \cdot \dfrac{\mathrm{P}_t(\boldsymbol{x})}{\mathrm{P}_s(\boldsymbol{x})}}
\end{aligned}
$$

$$= \frac{P_t(y=j \mid \boldsymbol{x}) \cdot \dfrac{P_s(\boldsymbol{x} \mid y=j)}{P_t(\boldsymbol{x} \mid y=j)} \cdot \dfrac{P_s(y=j)}{P_t(y=j)}}{\sum_{k \in [K]} P_t(y=k \mid \boldsymbol{x}) \cdot \dfrac{P_s(\boldsymbol{x} \mid y=k)}{P_t(\boldsymbol{x} \mid y=k)} \cdot \dfrac{P_s(y=k)}{P_t(y=k)}}$$

where $P_s$ is the probability distribution in the source domain (*i.e.,* the training dataset) and $P_t$ is the probability distribution in the target domain (*i.e.,* the test dataset). For long-tailed recognition, $P_s(y)$ appears a long-tailed distribution and $P_t(y)$ is a uniform distribution, *i.e.,* $P_t(y=k) = \dfrac{1}{K}$. Moreover, by assuming that $P_s(\boldsymbol{x} \mid y) = P_t(\boldsymbol{x} \mid y)$, we have

$$P_s(y=j \mid \boldsymbol{x}) = \frac{P_t(y=j \mid \boldsymbol{x}) \cdot P_s(y=j)}{\sum_{k \in [K]} P_t(y=k \mid \boldsymbol{x}) \cdot P_s(y=k)} \tag{7}$$

In deep classification models, $P_t(y \mid \boldsymbol{x})$ is estimated by the Softmax of the output logits $\boldsymbol{z}$, which is

$$P_t(y=j \mid \boldsymbol{x}) = \frac{\exp(z_j)}{\sum_{k \in [K]} \exp(z_k)} \tag{8}$$

In order to get the optimal probability model in the target domain, we substitute $P_t(y \mid \boldsymbol{x})$ in Eq. (7) with Eq. (8). In this way, we optimize the predicted probability in the source domain as

$$\begin{aligned}
P_s(y=j \mid \boldsymbol{x}) &= \frac{\dfrac{\exp(z_j)}{\sum_{k \in [K]} \exp(z_k)} \cdot P_s(y=j)}{\sum_{k \in [K]} \dfrac{\exp(z_k)}{\sum_{k' \in [K]} \exp(z'_k)} \cdot P_s(y=k)} \\
&= \frac{\exp(z_j) \cdot P_s(y=j)}{\sum_{k \in [K]} \exp(z_k) \cdot P_s(y=k)} \\
&= \frac{\exp(z_j + \log P_s(y=j))}{\sum_{k \in [K]} \exp(z_k + \log P_s(y=k))}
\end{aligned}$$

In practice, $P_s(y)$ can be estimated by calculating the class frequency in the training dataset. Compared to other elaborately designed losses, the logit-adjusted loss does not require any hyperparameters and has fewer assumptions about the data distribution, making it more generalizable across different backbone models. As shown in Table 10, some other elaborately designed losses such as LDAM and LADE, perform unsatisfactorily when applied to the CLIP model.

## C   EXPLANATION OF TEST-TIME ENSEMBLING

We present the detailed procedures of *test-time ensembling* (TTE) in Algorithm 1, using ViT-B/16 ($224 \times 224$ resolution) as the backbone model. The highlighted lines denote the additional steps introduced by TTE. Traditionally, an image is first resized and center-cropped, and then split into patches before being fed into the Transformer model. However, this conventional approach inevitably leads to the segmentation of important patterns across different patches, thus impeding the generalization. By employing diverse croppings, patterns that might be segmented in one cropping will be preserved in another. It is crucial to emphasize that the expanded size $e$ should not be a multiple of the patch size 16; otherwise, the five cropped images will share a significant portion of the same patches, rendering the expected diversity unattainable. In PEL, we default to set $e = 24$. Furthermore, we conduct a comparison of different expanded sizes and report the results in Figure 14.

Aside from TTE, we explore additional augmentation techniques such as TTE + Flipping and Random Augmentations (He et al., 2016) for multiple times. The results in Table 14 demonstrate that TTE is more effective than other augmentation methods.

## D   TEXTUAL PROMPTS FOR SEMANTIC-AWARE INITIALIZATION

In PEL, we use "`a photo of a [CLASS].`" as the template to generate textual prompts and then compute their features to initialize the classifier weights. One may be concerned with the impact

---

**Algorithm 1** TEST-TIME ENSEMBLING

---

**Input:** Image $x$, expanded size $e$, input resolution (224).

1: Resize $x$ to $x'$ sized $(224 + e) \times (224 + e)$.
2: Crop the center $224 \times 224$ portion of $x'$, denoted by $x^c$.
3: Split $x^c$ evenly into $m$ patches $[x_1^p; \cdots; x_m^p]$ (each $x_i^p$ is sized $16 \times 16$, and $m = \frac{224}{16} \times \frac{224}{16} = 196$).
4: Calculate the feature $f^c$ according to Section 3.1.
5: Calculate the logits $z^c$ according to Section 3.3.
6: Crop the top left $224 \times 224$ portion of $x'$, repeat procedure 3-5 and obtain the logits $z^{tl}$.
7: Crop the top right $224 \times 224$ portion of $x'$, repeat procedure 3-5 and obtain the logits $z^{tr}$.
8: Crop the bottom left $224 \times 224$ portion of $x'$, repeat procedure 3-5 and obtain the logits $z^{bl}$.
9: Crop the bottom right $224 \times 224$ of portion $x'$, repeat procedure 3-5 and obtain the logits $z^{br}$.

**Output:** Predicted logits $z = \text{Average}(z^c + z^{tl} + z^{tr} + z^{bl} + z^{br})$.

---

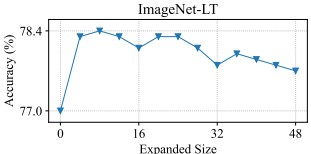 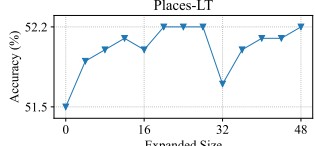 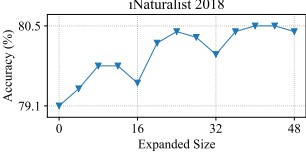

Figure 14: Performance of *test-time ensembling* (TTE) with different expanded size $e$. Setting $e = 0$ indicates not applying TTE. Setting $e$ to a multiple of the patch size 16 yields suboptimal performance. Generally, $e = 24$ is suitable for enhancing the generalization.

Table 14: Comparison of different augmentation methods on ImageNet-LT.

| Augmentation methods | Augmentation times | Overall | Many | Medium | Few |
|---|---|---|---|---|---|
| TTE (Ours) | 5 | **78.3** | **81.3** | **77.4** | **73.4** |
| TTE + Flipping | 10 | **78.3** | **81.3** | 77.3 | 73.3 |
| Random Augmentation | 5 | 76.7 | 79.7 | 75.5 | 71.9 |
| Random Augmentation | 10 | 77.3 | 80.3 | 76.3 | 72.2 |
| Random Augmentation | 15 | 77.7 | 80.8 | 76.6 | 72.3 |
| Random Augmentation | 20 | 77.8 | 80.9 | 76.7 | 72.7 |

of the used prompts. We conduct experiments to compare different prompting methods, including 1) the original class name ("[CLASS]") and 2) prompt ensembling (Radford et al., 2021) which applies different templates to class names. The results in Table 15 show that these prompts have a similar performance, and using "a photo of a [CLASS]" is adequate for generalization.

Moreover, we posit that CLIP has seen sufficient language corpus, considering its pre-training on web-scale datasets. However, it is noteworthy that CLIP may fail to recognize specific class names. This probably stems from its limited vocabulary size (CLIP contains approximately 49K vocabulary) or encountering uncommon or novel concepts. In this case, semantic-aware initialization may regress to random initialization. In response to this, we explore an alternative approach by incorporating class descriptions (which can be crafted manually or generated using large language models). In practice, we follow Menon & Vondrick (2023) to generate the descriptions for each class, then combine these descriptions and calculate the textual feature for initialization. The results are reported in the bottom line of Table 15, which shows that the use of class descriptions can also enhance the performance compared to random initialization.

## E  MODEL PARAMETER QUANTITY ANALYSIS

The components of the CLIP Vision Transformer (CLIP-ViT) are detailed in Table 17. CLIP-ViT comprises an Embedding layer and $L$ Transformer blocks. While there may be various versions of pre-trained Vision Transformers, the differences between them are generally minor. The total number of parameters in CLIP-ViT can be computed as follows: $(12L + 1)d^2 + (13L + m + 6)d$, where $d$ represents the dimension of the embedding features, and $m$ denotes the number of image

Table 15: Comparison of different prompting methods on ImageNet-LT.

| Prompting methods | Overall | Many | Medium | Few |
|---|---|---|---|---|
| None prompt (random initialization) | 76.1 | 80.8 | 75.9 | 63.2 |
| "[CLASS]" | 78.2 | **81.4** | 77.3 | 72.3 |
| "A photo of a [CLASS]" | **78.3** | 81.3 | **77.4** | **73.4** |
| Prompt ensembling | **78.3** | 81.3 | **77.4** | 73.3 |
| Class descriptions (w/o class names) | 77.4 | 81.3 | 76.9 | 68.2 |

patches. For instance, in the case of ViT-B/16, where $L = 12$, $d = 768$, $m = 196$, the parameter quantity amounts to 85,799,424 ($\approx$ 85.80M).

In addition, we present the parameter quantities of the PEFT modules in Table 18 (for detailed definitions, please refer to Section 3.3). The table illustrates that for all PEFT modules, the parameter quantities are at the polynomial level of $d$. Notably, $p$ for VPT and $r$ for Adapter are manually set hyperparameters and are much smaller than $d$. In comparison to the entire Transformer block, a PEFT module is significantly more lightweight.

Moreover, the parameter quantity for a classifier is approximately $Kd$, where $K$ is the number of classes. In PEL, we set the bottleneck dimension $r = 2^{\lfloor \log_2 (\frac{K}{2L}) \rfloor} \leq \frac{K}{2L}$ for the AdaptFormer, so that the total parameter quantity is $L \cdot 2rd \leq Kd$ (ignoring constant terms). As a result, it learns even fewer parameters than the classifier.

We also record the time cost of PEL with each PEFT method and report the results in Table 16. The results suggest that the time costs for different PEFT methods are highly close.

Table 16: Training time per epoch when using different PEFT methods.

| Methods | | ImageNet-LT | Places-LT |
|---|---|---|---|
| | LN tuning | 2 min 49 s | 1 min 30 s |
| | BitFit | 2 min 51 s | 1 min 33 s |
| | VPT-shallow | 2 min 46 s | 1 min 31 s |
| PEL w/ | VPT-deep | 2 min 56 s | 1 min 37 s |
| | Adapter | 2 min 49 s | 1 min 33 s |
| | LoRA | 2 min 58 s | 1 min 31 s |
| | AdaptFormer | 2 min 58 s | 1 min 38 s |

Table 17: Model architecture and parameter quantity for CLIP-ViT.

| Layers | Components | Variables | Size | #Params. |
|---|---|---|---|---|
| Embedding | Projection | - | $d \times d$ | $d^2 + (m+2)d$ |
| | Class Token | $\boldsymbol{c}^0$ | $d$ | |
| | Positional | - | $(m+1) \times d$ | |
| | LN | $\boldsymbol{\gamma}, \boldsymbol{\beta}$ | $d, d$ | $2d$ |
| Block-1 $(l=1)$ | LN | $\boldsymbol{\gamma}, \boldsymbol{\beta}$ | $d, d$ | $12d^2 + 13d$ |
| | MSA | $\{\boldsymbol{W}_Q^{l,h}, \boldsymbol{b}_Q^{l,h}\}_{h=1}^H$ | $\{d \times \frac{d}{H}, \frac{d}{H}\} \times H$ | |
| | | $\{\boldsymbol{W}_K^{l,h}, \boldsymbol{b}_K^{l,h}\}_{h=1}^H$ | $\{d \times \frac{d}{H}, \frac{d}{H}\} \times H$ | |
| | | $\{\boldsymbol{W}_V^{l,h}, \boldsymbol{b}_V^{l,h}\}_{h=1}^H$ | $\{d \times \frac{d}{H}, \frac{d}{H}\} \times H$ | |
| | | $\boldsymbol{W}_O^l, \boldsymbol{b}_O^l$ | $d \times d, d$ | |
| | LN | $\boldsymbol{\gamma}, \boldsymbol{\beta}$ | $d, d$ | |
| | FFN | $\boldsymbol{W}_1^l, \boldsymbol{b}_1^l$ | $d \times 4d, 4d$ | |
| | | $\boldsymbol{W}_2^l, \boldsymbol{b}_2^l$ | $4d \times d, d$ | |
| $\vdots$ | $\vdots$ | $\vdots$ | $\vdots$ | $\vdots$ |
| Block-$L$ | $\cdots$ | $\cdots$ | $\cdots$ | $12d^2 + 13d$ |
| | LN | $\boldsymbol{\gamma}, \boldsymbol{\beta}$ | $d, d$ | $2d$ |

Table 18: Parameter quantities for different PEFT modules in a Transformer block.

| Modules | Components | Variables | Size | #Params. |
|---|---|---|---|---|
| LN tuning | LN | $\boldsymbol{\gamma}, \boldsymbol{\beta}$ | $d, d$ | $4d$ |
| | LN | $\boldsymbol{\gamma}, \boldsymbol{\beta}$ | $d, d$ | |
| BitFit | LN-bias | $\boldsymbol{\beta}$ | $d$ | $11d$ |
| | MSA-bias | $\{\boldsymbol{b}_Q^{l,h}\}_{h=1}^H$ | $\{\frac{d}{H}\} \times H$ | |
| | | $\{\boldsymbol{b}_K^{l,h}\}_{h=1}^H$ | $\{\frac{d}{H}\} \times H$ | |
| | | $\{\boldsymbol{b}_V^{l,h}\}_{h=1}^H$ | $\{\frac{d}{H}\} \times H$ | |
| | | $\boldsymbol{b}_O^l$ | $d$ | |
| | LN-bias | $\boldsymbol{\beta}$ | $d$ | |
| | FFN-bias | $\boldsymbol{b}_1^l$ | $4d$ | |
| | | $\boldsymbol{b}_2^l$ | $d$ | |
| VPT | Prompts | $\boldsymbol{P}^l$ | $p \times d$ | $pd$ |
| Adapter | LN | $\boldsymbol{\gamma}, \boldsymbol{\beta}$ | $d, d$ | $(2r+3)d + r$ |
| | Projection | $\boldsymbol{W}_{\text{down}}, \boldsymbol{b}_{\text{down}}$ | $d \times r, r$ | |
| | | $\boldsymbol{W}_{\text{up}}, \boldsymbol{b}_{\text{up}}$ | $r \times d, d$ | |
| LoRA | Projection | $\boldsymbol{W}_{\text{down}}, \boldsymbol{W}_{\text{up}}$ (for $\boldsymbol{W}_Q$) | $d \times r, r \times d$ | $4rd$ |
| | | $\boldsymbol{W}_{\text{down}}, \boldsymbol{W}_{\text{up}}$ (for $\boldsymbol{W}_V$) | $d \times r, r \times d$ | |
| AdaptFormer | LN | $\boldsymbol{\gamma}, \boldsymbol{\beta}$ | $d, d$ | $(2r+3)d + r + 1$ |
| | Projection | $\boldsymbol{W}_{\text{down}}, \boldsymbol{b}_{\text{down}}$ | $d \times r, r$ | |
| | | $\boldsymbol{W}_{\text{up}}, \boldsymbol{b}_{\text{up}}$ | $r \times d, d$ | |
| | Scaling | $s$ | $1$ | |

