# OpenReview forum: "Parameter-Efficient Long-Tailed Recognition"
_ICLR.cc/2024/Conference — Submitted to ICLR 2024_

### Official Review · Reviewer_4hL6 · 2023-10-28

**Soundness:** 3 good
**Presentation:** 3 good
**Contribution:** 2 fair
**Rating:** 5
**Confidence:** 4

**Summary:**

This paper introduces PEL, a parameter-efficient finetuning method for long-tailed recognition. It firstly analyzes the overfitting issue caused by full finetuning or classifier finetuning. Then it adopts several techniques to mitigate the problem, including:

- using existing Parameter-Efficient Fine-Tuning (PEFT) methods (e.g. AdaptFormer) instead;

- initializing the classifier weights with classwise textual features generated by CLIP;

- ensembling logits from different cropping methods of the same image during test time.


Based on these, this paper achieves SoTA results on several common long-tailed benchmarks with fewer training epochs and parameters.

**Strengths:**

- The paper is well-written and easy to follow.

- The experiments results are strong, and the ablations are clear and detailed.

- Different PEFT methods are explored for the long-tailed recognition (LTR) task.

**Weaknesses:**

- The technical novelty is a little incremental. All these PEFT methods already exist, and the authors adopt them for LTR tasks directly without any tailored modification. The initialization and ensembling are more like some kind of tricks without much insights for the community.

- Many previous works adopt ResNet as backbone. However, no experment in this paper adopts ResNet as well, which may raise questions about its generalisability.

**Questions:**

1. Why only use image cropping for test-time ensembling? Have you tried other data augmentation methods?

2. What kind of textual prompts are used for semantic-aware initialization?

---

> ### Author Response · Authors · 2023-11-14
>
> We are grateful for the thoughtful reviews, and for commenting that the paper is *well-written* and the experiment results are *strong*. We will address the concerns below.
>
> **Concern #1 (Weakness #1): The technical novelty is a little incremental.**
>
> **Response:** Our work pioneers a systematical study on how to efficiently and effectively adapt pre-trained models for long-tailed tasks. Throughout our investigation, we have discovered many new insights, and we aim to **disclose these fundamental findings** and **inspire future research**.
>
> 1) Full Fine-tuning fails in long-tailed learning, and PEFT emerges as a superior alternative. This contradicts common sense on balanced datasets, where PEFT typically strives to match the performance of full fine-tuning [1-5].
> 2) We discover that most elaborately designed losses fail when applied to CLIP, except Logit-Adjusted loss (explained in Appendix B). This discovery motivates future research to design more suitable losses.
> 3) We point out the importance and advantages of classifier initialization, particularly for tail classes. This also inspires more advanced research in the future.
> 4) We first propose to employ TTE to remedy the inherent limitations of Vision Transformer. As a remedial measure, TTE is intuitive and effective (explained in Appendix C).
>
> Moreover, the components we have introduced are both scalable and effective. We believe our findings and efforts can contribute to the long-tailed learning community.
>
>
>
> **Concern #2: (Weakness #2): Adopt ResNet as backbone.**
>
> **Response:** We have also considered this problem. However, there is no PEFT method specifically designed for ResNet, making it challenging to integrate our method with ResNet. Despite this constraint, we have explored some straightforward strategies, including 1) incorporating a scaling and shifting (SSF) [6] module after the backbone, and 2) fine-tuning solely the bias terms of ResNet. The results are presented below. In comparison to zero-shot CLIP and previous methods (detailed in Tables 1&2 in the main paper), our method achieves significantly superior performance with lower computational costs.
>
> - ImageNet-LT
>
>   |                          | Backbone  | Learnable Params. | #Epochs | Overall  | Many     | Medium   | Few      |
>   | ------------------------ | --------- | ----------------- | ------- | -------- | -------- | -------- | -------- |
>   | Zero-shot CLIP           | ResNet-50 | -                 | -       | 57.6     | 58.6     | 56.9     | 56.9     |
>   | PEL w/ SSF               | ResNet-50 | 0.002M            | 10      | 66.9     | 72.0     | 66.5     | 54.1     |
>   | PEL w/ bias tuning       | ResNet-50 | 0.034M            | 10      | 67.8     | 72.0     | 67.3     | 57.6     |
>   | PEL w/ bias tuning & SSF | ResNet-50 | 0.036M            | 10      | **68.3** | **72.5** | **67.8** | **58.2** |
>
> - Places-LT
>
>   |                          | Backbone  | Learnable Params. | #Epochs | Overall  | Many     | Medium   | Few      |
>   | ------------------------ | --------- | ----------------- | ------- | -------- | -------- | -------- | -------- |
>   | Zero-shot CLIP           | ResNet-50 | -                 | -       | 35.2     | 33.1     | 34.6     | 40.4     |
>   | PEL w/ SSF               | ResNet-50 | 0.002M            | 10      | 46.7     | 47.5     | 48.7     | 40.7     |
>   | PEL w/ bias tuning       | ResNet-50 | 0.034M            | 10      | 47.9     | **48.2** | 49.8     | 42.9     |
>   | PEL w/ bias tuning & SSF | ResNet-50 | 0.036M            | 10      | **48.1** | 48.1     | **50.0** | **43.9** |
>
>
>
> [1] BitFit: Simple Parameter-efficient Fine-tuning for Transformer-based Masked Language-models.
>
> [2] Visual Prompt Tuning.
>
> [3] Parameter-Efficient Transfer Learning for NLP.
>
> [4] LoRA: Low-rank adaptation of large language models
>
> [5] AdaptFormer: Adapting Vision Transformers for Scalable Visual Recognition.
>
> [6] Scaling & Shifting Your Features: A New Baseline for Efficient Model Tuning.

---

> ### Author Response · Authors · 2023-11-14
>
> **Concern #3 (Question #1): Why only use image cropping for test-time ensembling? Have you tried other data augmentation methods?**
>
> **Response:** We utilize image cropping tailored for Vision Transformer (ViT), which also uses image cropping to obtain $n\times n$ image patches. ViT carries the inherent risk of pattern separation. In response, we intuitively generate diverse cropped patches to remedy this issue. The detailed explanations are given in Appendix C. The results in Figure 12 demonstrate the effectiveness of our method: 1) When our cropped patches align with those of ViT, pattern separation persists, and the performance improvement is limited; 2) Conversely, when our cropped patches differ from ViT, the pattern separation can be alleviated, and the performance is enhanced.
>
> Aside from TTE, we explore additional augmentation techniques such as TTE + Flipping and Random Augmentations [6] for multiple times. The results below demonstrate that TTE is more effective than other methods.
>
> | Augmentation methods | Augmentations times | Overall  | Many     | Medium   | Few      |
> | -------------------- | ------------------- | -------- | -------- | -------- | -------- |
> | TTE (Ours)           | 5                   | **78.3** | **81.3** | **77.4** | **73.4** |
> | TTE + Flipping       | 10                  | **78.3** | **81.3** | 77.3     | 73.3     |
> | Random Augmentation  | 5                   | 76.7     | 79.7     | 75.5     | 71.9     |
> | Random Augmentation  | 10                  | 77.3     | 80.3     | 76.3     | 72.2     |
> | Random Augmentation  | 15                  | 77.7     | 80.8     | 76.6     | 72.3     |
> | Random Augmentation  | 20                  | 77.8     | 80.9     | 76.7     | 72.7     |
>
>
>
> **Concern #4 (Question #2): What kind of textual prompts are used for semantic-aware initialization?**
>
> **Response:** We employ the most widely used textual prompt, "a photo of a [CLASS]", where [CLASS] can be replaced by the class names. Additionally, we compare different prompting methods, including 1) the original class name ("[CLASS]") and 2) prompt ensembling [7] which applies different templates to class names. The experiments are conducted on ImageNet-LT. The results below show that these prompts have similar performance, and using "a photo of a [CLASS]" is adequate for generalization.
>
> Moreover, Reviewer BCXb has proposed an intriguing situation, i.e., when the class names are OOD or unseen for CLIP. In response to this, we explore an alternative approach by incorporating class descriptions (which can be crafted manually or generated using large language models). The results show that using class descriptions can also enhance the performance compared to random initialization.
>
> | Prompting methods                    | Overall  | Many     | Medium   | Few      |
> | ------------------------------------ | -------- | -------- | -------- | -------- |
> | Random initialization                | 76.1     | 80.8     | 75.9     | 63.2     |
> | "[CLASS]"                            | 78.2     | **81.4** | 77.3     | 72.3     |
> | "a photo of a [CLASS]"               | **78.3** | 81.3     | **77.4** | **73.4** |
> | Prompt ensembling                    | **78.3** | 81.3     | **77.4** | 73.3     |
> | Class descriptions (w/o class names) | 77.4     | 81.3     | 76.9     | 68.2     |
>
>
>
> [6] Deep residual learning for image recognition.
>
> [7] Learning Transferable Visual Models From Natural Language Supervision.

---

> > ### Comment · Reviewer_4hL6 · 2023-11-22
> >
> > Thank you for the detailed feedback and additional experiments. I have read all the comments from both authors and other reviewers.
> >
> > > Our work pioneers a systematical study on how to efficiently and effectively adapt pre-trained models for long-tailed tasks.
> >
> > I agree that this paper provides a systematical study on different PEFT methods for long-tailed recognition (LTR), as is shown in Tab. 4 and Tab. 5. However, those new empirical insights listed in the rebuttall are not aligned with this paper’s main claim. The paper originally intends to propose a new framework for LTR, rather than disclosing those empirical findings for existing methods.
> >
> > > **Concern #2: (Weakness #2): Adopt ResNet as backbone.**
> >
> > From my perspective, this concern is not well addressed in the rebuttal. The authors try to use other PEFT methods (such as SSF and bias tuning) rather than transferring the original PEL they adopt for ViT to ResNet. Although the performance is promising, it cannot serve as the evidence for this paper’s main contribution.
> >
> > > ViT carries the inherent risk of pattern separation.
> >
> > Pattern separation does not necessarily induce any risk for ViT, as we can apply strong data augmentations such as RandomResizeCrop during training to mitigate the spatial inductive bias caused by Patchifying. I am still wondering why only using Random Cropping is better than further ensembling other data augmentation techniques.
> >
> >
> > In summary, I agree with Reviewer BCXb and Reviewer GTCk that the writing of this paper needs to be further refined, as it now looks more like an empirical study of using different backbone-dependent PEFT methods for long-tailed learning.

---

> > > ### Author Response · Authors · 2023-11-22
> > >
> > > Dear Reviewer,
> > >
> > > We would like to appreciate your feedback. There might be some misunderstandings regarding your concerns, and we will make the clarifications below.
> > >
> > > **Concern: Those new empirical insights listed in the rebuttal are not aligned with this paper’s main claim.**
> > >
> > > **Response:** Thank you for pointing out this. As we also responded to Reviewer GTCk and BCXb, **the main contributions of this paper lie in the proposed framework and the state-of-the-art performance it achieves**.
> > >
> > > However, the proposed framework raises the concerns of novelty. Therefore, we argue that our work reveals several insights. And these insights are also important to contribute to the field.
> > >
> > >
> > >
> > > **Concern: The authors try to use other PEFT methods (such as SSF and bias tuning) rather than transferring the original PEL they adopt for ViT to ResNet.  Although the performance is promising, it cannot serve as the evidence for this paper’s main contribution.**
> > >
> > > **Response:** There might be some misunderstandings.
> > >
> > > First, why do not we transfer the original PEFT methods we adopt for ViT? **Because they are not designed for ResNet.**
> > >
> > > - LN tuning adjusts the LayerNorm modules of ViT. However, ResNet does not contain LayerNorm modules.
> > > - VPT-shallow and VPT-deep add learnable prompts into ViT. However, ResNet does not contain prompts.
> > >
> > > - LoRA is integrated into the *Multi-Head Self-Attention* layer of ViT. However, ResNet does not contain this layer.
> > > - Adapter and AdaptFormer are integrated into the *Feed-Forward Network* layer of ViT. However, ResNet does not contain this layer.
> > >
> > > - BitFit fine-tunes only the bias parts of ViT. This idea can be adapted to ResNet. Hence, we explore "bias tuning" for ResNet.
> > >
> > > Moreover, there is no PEFT method proposed for ResNet. We have referred to the recent survey [1]. Except for full fine-tuning, ResNet cannot adopt any PEFT methods.
> > >
> > > Second, **we have never claimed that PEL with ResNet as the backbone is a main contribution.** Our paper is mainly based on Transformer-based models.
> > >
> > >
> > >
> > > **Concern: Pattern separation does not necessarily induce any risk for ViT, as we can apply strong data augmentations such as RandomResizeCrop during training to mitigate the spatial inductive bias caused by Patchifying. I am still wondering why only using Random Cropping is better than further ensembling other data augmentation techniques.**
> > >
> > > **Response:** There might be some misunderstandings.
> > >
> > > First, **we have already applied RandomResizedCrop during training** (please refer to our source code and check line 98 of `trainer.py`). Nonetheless, adopting TTE still boosts the performance.
> > >
> > > Second, **TTE is not Random Cropping.** It has a deterministic process. Please refer to Algorithm 1 and Appendix D for detailed explanations. Moreover, TTE performs better than Random Cropping, as is compared in Table 14.
> > >
> > >
> > >
> > > [1] Visual Tuning. arXiv preprint arXiv:2305.06061.

---

### Official Review · Reviewer_GTCk · 2023-10-30

**Soundness:** 2 fair
**Presentation:** 3 good
**Contribution:** 2 fair
**Rating:** 5
**Confidence:** 4

**Summary:**

In the study, the authors introduce PEFT (Parameter Efficient Fine-Tuning), a method designed for efficient adaptation of pre-trained models for long-tailed recognition challenges, achieving this adaptation in a reduced number of epochs and without additional data requirements. In this paper, PEL is also proposed to introduce a small number of task-specific parameters and present a novel semantic-aware classifier initialization strategy. Experimental results demonstrate the enhanced performance of the algorithm.

**Strengths:**

- The topic of PEFT (Parameter Efficient Fine-Tuning) is important, enabling a low-cost adaptation for downstream applications.
- This paper conducts empirical comparisons with different paradigms on long-tailed benchmark, such as training from scratch, finetune pretrained model and finetune with additional data.
- The proposed method is effective without introducing many computational cost and extra data.
- Generally, the paper is well structured and easy to follow.

**Weaknesses:**

- The related work seems insufficient. The authors should include references to existing studies concerning parameter-efficient fine-tuning approaches. Although some PEFT methods are presented in Section 3.3, these methods are listed without detailed explanations.
- The technical contribution of this paper is not clear. The choice of AdaptFormer, based primarily on its state-of-the-art results, appears unrelated to long-tailed learning and should not be considered a contribution to the field. Besides, the initialization from CLIP text embeddings can not be regarded as significant technical contribution. Logit-adjusted (LA) loss and test-time ensembling also has been proposed.
- In Table 4, the authors seem only conduct experiments with existing PEL methods, the performance of the proposed method is missing.
- The authors should compare the computational cost with PEL methods in Table 4, instead of only comparing with non-PEL methods.
- The paper should broaden its scope by comparing various long-tailed learning strategies, rather than focusing on results from Logit adjustment or cosine classifer. The current benchmark could undermine the generalizability and confidence of the results.

**Questions:**

- The paper lacks intuitive explanations. The authors claim that their method can prevent overfitting and decrease training epochs without extra data, but how is PEFT helping? More empirical or theoretical explanations are suggested.
- Why are the training epochs chosen as 10/20? How would the results differ if the training period were extended?
- In Table 3, PEL incurs larger(4x) computation cost than LPT. Besides, LPT is not compared in Table 1. Will PEL degenerate on more real-world long-tailed datasets? It is suggested to add more datasets to enhance the confidence of the results.

**Details Of Ethics Concerns:**

No.

---

> ### Author Response · Authors · 2023-11-14
>
> We appreciate the reviewer for the thorough comments, and for mentioning that the topic is *important* and the paper is *well structured*. Following we will respond to the concerns.
>
> **Concern #1 (Weakness #1): Include references and detailed explanations of existing studies concerning parameter-efficient fine-tuning approaches.**
>
> **Response:** This is a good suggestion. We will add more explanations as follows:
>
> Parameter-efficient fine-tuning (PEFT) aims to adapt the pre-trained models to downstream tasks by learning a small number of parameters. Over the past few years, various PEFT approaches have been proposed for both computer vision (CV) and natural language processing (NLP) tasks [1]. These approaches can be broadly categorized into three groups [1]: 1) Prompt Tuning, such as VPT-shallow [3] and VPT-deep [3], is inspired by prompting strategies in NLP. It prepends learnable patches to images or sentences to serve as the contexts for downstream tasks; 2) Adapter Tuning, such as Adapter [4] and AdaptFormer [6], incorporates lightweight modules sequentially or parallelly to the pre-trained models; 3) Parameter Tuning, like BitFit [2] and LoRA [5], selectively modifies specific parts of the pre-trained models, such as the weight part or the bias part. Recent work [7] offers a unified view of different PEFT methods and proposes that they are intrinsically similar. Nevertheless, different PEFT modules may excel in distinct tasks due to variations in their design approaches and integration positions.
>
> Thanks for the thoughtful comments. We will update the related works in the next version.
>
>
>
> **Concern #2 (Weakness #2): The technical contribution of this paper is not clear.**
>
> **Response:** We summarize the contribution of this work as follows.
>
> PEL is a general framework, with all its components scalable and efficient. Its design allows updates in alignment with the development of the related field. What is more, we aim to **reveal fundamental insights** and **inspire future research** regarding long-tailed learning with pre-trained models.
>
> 1) Full Fine-tuning fails in long-tailed learning, and PEFT emerges as a superior alternative. This contradicts common sense on balanced datasets, where PEFT typically strives to match the performance of full fine-tuning [1-5].
> 2) We discover that most elaborately designed losses fail when applied to CLIP, except Logit-Adjusted loss (explained in Appendix B). This discovery motivates future research to design more suitable losses.
> 3) We point out the importance and advantages of classifier initialization, particularly for tail classes. This also inspires more advanced research in the future.
> 4) We first propose to employ TTE to remedy the inherent limitations of Vision Transformer. As a remedial measure, TTE is intuitive and effective (explained in Appendix C).
>
> To the best of our knowledge, this is the first work that systematically studies the parameter-efficient adaptation of pre-trained models on long-tailed tasks. We believe our findings and efforts can contribute to the long-tailed learning community.
>
>
>
> **Concern #3 (Weakness #3): In Table 4, the authors seem only conduct experiments with existing PEL methods, the performance of the proposed method is missing.**
>
> **Response:** There may be some misunderstandings. PEL is a framework that can adopt any existing PEFT methods. Table 4 is an ablation study on different PEFT methods. The results show that PEL with AdaptFormer achieves superior performance currently. If some new PEFT methods are proposed in the future, PEL can seamlessly incorporate them to enhance its performance.
>
> Moreover, we apologize for the confusion in Table 4. It should be clarified that zero-shot CLIP, full fine-tuning, and classifier fine-tuning are not PEFT methods. The remaining methods are PEFT methods that can be integrated into PEL. We will update Table 4 in the next version.
>
>
>
>
> [1] Visual Tuning.
>
> [2] BitFit: Simple Parameter-efficient Fine-tuning for Transformer-based Masked Language-models.
>
> [3] Visual Prompt Tuning.
>
> [4] Parameter-Efficient Transfer Learning for NLP.
>
> [5] LoRA: Low-rank adaptation of large language models
>
> [6] AdaptFormer: Adapting Vision Transformers for Scalable Visual Recognition.
>
> [7] Towards a Unified View of Parameter-Efficient Transfer Learning.

---

> ### Author Response · Authors · 2023-11-14
>
> **Concern #4 (Weakness #4): Compare the computational cost with PEL methods.**
>
> **Response:** Thank you for the suggestion. We compute the time cost per epoch for different parameter-efficient fine-tuning methods. The experiments are conducted using a single NVIDIA A100 GPU. The results indicate that the time cost for each PEFT method is very close.
>
> | Methods     | ImageNet-LT | Places-LT  |
> | ----------- | ----------- | ---------- |
> | LN tuning   | 2 min 49 s  | 1 min 30 s |
> | BitFit      | 2 min 51 s  | 1 min 33 s |
> | VPT-shallow | 2 min 46 s  | 1 min 31 s |
> | VPT-deep    | 2 min 56 s  | 1 min 37 s |
> | Adapter     | 2 min 49 s  | 1 min 33 s |
> | LoRA        | 2 min 58 s  | 1 min 31 s |
> | AdaptFormer | 2 min 58 s  | 1 min 38 s |
>
>
>
> **Concern #5 (Weakness #5): Comparing various long-tailed learning strategies.**
>
> **Response:** We have compared different classifiers and losses in Tables 9 and 10 in Appendix A. Moreover, we give an analysis of Logit Adjusted loss in Appendix B. Please refer to the appendix for detailed explanations.
>
> To enhance clarity, we will add a concise outline in the paper to summarize the experiments conducted in the appendix.
>
>
>
> **Concern #6 (Question #1): How is PEFT helping?**
>
> **Response:** As illustrated in Figure 4 in the main paper, the utilization of PEFT enhances the capability to extract more separable representations. This is because the PEFT module is integrated into the backbone, and enhances the feature extraction process of the pre-trained model. In this way, it is more effective than solely fine-tuning the classifier.
>
> Moreover, PEFT freezes the pre-trained model while introducing a small number of learnable parameters for adaptation. This approach ensures the retention of the generalization ability of the pre-trained model. In contrast to full fine-tuning, PEFT can mitigate overfitting, as analyzed in Table 8 in Appendix A.
>
>
>
> **Concern #7 (Question #2): Why are the training epochs chosen as 10/20? How would the results differ if the training period were extended?**
>
> **Response:** We set the training epochs to 10 for ImageNet-LT, Places-LT, and CIFAR-100-LT. This is because PEL can converge rapidly by learning a small number of parameters, along with a classifier initialization.
>
> For iNaturalist 2018, we train 20 epochs considering it has much more data. We have presented an analysis of different training epochs in Table 7 in Appendix A. The results show that extending the training epochs can result in improved performance. However, this improvement comes at the expense of increased computational overhead.
>
> We also conduct experiments across different epochs on ImageNet-LT and Places-LT, and the results are presented below. Increasing the training epochs does not yield significant improvements. In practice, the selection of the optimal training epoch depends on the data scale and the task complexity. Generally, 10-20 epochs are appropriate for most cases.
>
> - Results of PEL on ImageNet-LT by training different epochs.
>
>   | #Epochs | Overall  | Many | Medium   | Few      |
>   | ------- | -------- | ---- | -------- | -------- |
>   | 5       | 77.5     | 80.9 | 77.2     | 69.0     |
>   | 10      | **78.3** | 81.3 | **77.4** | **73.4** |
>   | 20      | **78.3** | 81.8 | 77.1     | 72.8     |
>   | 30      | 78.0     | 82.2 | 76.6     | 71.1     |
>
> - Results of PEL on Places-LT by training different epochs.
>
>   | #Epochs | Overall  | Many     | Medium   | Few      |
>   | ------- | -------- | -------- | -------- | -------- |
>   | 5       | 51.3     | **51.7** | 53.0     | 46.8     |
>   | 10      | **52.2** | **51.7** | **53.1** | 50.9     |
>   | 20      | 51.8     | 51.3     | 52.3     | **51.6** |
>   | 30      | 51.3     | 51.4     | 51.5     | 50.5     |
>
>
>
> **Concern #8 (Question #3): PEL incurs larger(4x) computation cost than LPT.**
>
> **Response:** We have discussed this in Paragraph *Results on iNaturalist 2018* in Section 4.2: Although LPT uses fewer learnable parameters, we can reduce the parameters of PEL to reach a comparable quantity (i.e., reduce the bottleneck dimension r to 64, more details are given in Figure 6). In this case, PEL achieves an accuracy of 77.7% (without TTE) / 79.0% (with TTE), which still outperforms LPT. In fact, due to the large number of classes of iNaturalist 2018, the classifier already contains 6.25M parameters. Therefore, the parameter quantity of PEL does not lead to too much cost.

---

> > ### Comment · Reviewer_GTCk · 2023-11-22
> >
> > Thanks for the response. I have read the response and other reviews.
> >
> > A major concern is the technical contribution, as also noted by Reviewer BCXb and Reviewer 4hL6. This concern isn't adequately addressed by merely outlining empirical findings and their broader implications.
> >
> > > e.g. We discover that most elaborately designed losses fail when applied to CLIP, except Logit-Adjusted loss (explained in Appendix B). This discovery motivates future research to design more suitable losses)
> >
> > Specially,  I agree with Reviewer BCXb that this paper necessitates a major revision, particularly if it aims to be a systematical empirical evaluations for the parameter-efficient finetuning in long-tailed learning scenarios. The current version lacks an in-depth analysis and a clear understanding of how different PEFT methods benefit long-tailed learning. Furthermore, the main body of the experimental part (Table 1-3) seems inadequate as it appears to focus more on confirming SOTA performance, rather than providing a comprehensive evaluation of various PEFT methods.
> >
> > > In practice, the selection of the optimal training epoch depends on the data scale and the task complexity.
> >
> > It seems PEL is sensitive to the training epochs. However, it remains unclear how to best determine the length of training epochs by data scale and the task complexity.

---

> ### Author Response · Authors · 2023-11-22
>
> Dear Reviewer,
>
> We appreciate you for your further feedback. We would like to address your concerns below.
>
> **Concern: A major concern is the technical contribution, as also noted by Reviewer BCXb and Reviewer 4hL6. This concern isn't adequately addressed by merely outlining empirical findings and their broader implications.**
>
> **Response:** First, as we further responded to Reviewer BCXb, **the main contributions of this paper lie in the proposed framework and the state-of-the-art performance it achieves**. Our method achieves significant performance improvements, which can help take an important stride forward in the field of long-tailed recognition.
>
> Second, **the technical contribution does not necessarily mean proposing a completely new technology**. Applying classic methods to specific fields, as well as proposing uncovering empirical findings that have not been previously explored, can also help advance the field. Numerous works are built upon this approach, and the following are just some typical examples:
>
> - Kang et al. [1] explore the widely used two-stage learning framework to help long-tailed recognition.
> - Yang et al. [2] study the effects of self-supervised learning and semi-supervised learning on long-tailed recognition.
> - Zhong et al. [3] employ the popular mixup method and label smoothing to help long-tailed representation learning.
> - Menon et al. [4] revisit the classic idea of logit adjustment based on label frequencies and use it to help long-tailed learning.
> - Park et al. [5] chose the CutMix technology to improve the data augmentation in long-tailed recognition.
> - Dong et al. [6] utilize visual prompt tuning, and previously proposed A-GCL loss to improve long-tailed recognition.
>
> All of these works have not proposed new technologies. Instead, they proposed a framework that utilizes classical methods to enhance long-tailed recognition tasks. However, they all provide novel insights into the field and have inspired several works.
>
> [1] Decoupling Representation and Classifier for Long-Tailed Recognition. ICLR 2020
>
> [2] Rethinking the Value of Labels for Improving Class-Imbalanced Learning. NeurIPS 2020.
>
> [3] Improving Calibration for Long-Tailed Recognition. CVPR 2021.
>
> [4] Long-Tail Learning via Logit Adjustment. ICLR 2021.
>
> [5] The Majority Can Help The Minority: Context-rich Minority Oversampling for Long-tailed Classification. CVPR 2022.
>
> [6] LPT: Long-tailed Prompt Tuning for Image Classification. ICLR 2023.
>
>
>
> **Concern: The current version lacks an in-depth analysis and a clear understanding of how different PEFT methods benefit long-tailed learning. Furthermore, the main body of the experimental part (Table 1-3) seems inadequate as it appears to focus more on confirming SOTA performance, rather than providing a comprehensive evaluation of various PEFT methods.**
>
> **Response:** In fact, we have already conducted plenty of work to analyze these PEFT methods in our paper.
>
> 1. We have analyzed the impact of PEFT modules on the representation in Figure 4 in Section 4.3.
> 2. We have analyzed the effect of the PEFT module on different layers in Figure 8 in Appendix A.
> 3. We have analyzed the influence of the PEFT module on model convergence in Figures 9 and 10 in Appendix A.
> 4. We have provided model parameter quantity analysis in Appendix E, and demonstrate that these PEFT methods are at the same complexity level and are all lightweight.
> 5. We have compared different PEFT methods in Table 4 in Section 4.3, and find that most of them yield enhanced performance.
>
> Moreover, we have updated our manuscript and **added Figures 12 and 13** to give more analyses on the representation separability of different PEFT methods. The results show that all of these PEFT methods contribute to more distinctive features. Notably, Adapter and AdaptFormer are more effective in enhancing separability.
>
> We would like to argue that, **our paper is not a simple "empirical evaluation"**. We have presented comprehensive analyses of the effects of the employed PEFT modules in multiple aspects.
>
>
>
> **Concern: It seems PEL is sensitive to the training epochs. However, it remains unclear how to best determine the length of training epochs by data scale and the task complexity.**
>
> **Response:** It is necessary to mention that, previous works also need to elaborately choose a proper training epoch. For instance, their training epochs range from 90 to 400 on ImageNet-LT, and 30 to 100 on Places-LT (please refer to Tables 1 and 2 for more details). However, none of the previous works have explained why they chose a specific number of training epochs.
>
> Moreover, the number of the training epochs is the only hyperparameter that needs to be manually set in our method. As we explained in the previous comments, training for 10-20 epochs is appropriate for PEL in most cases.

---

### Official Review · Reviewer_egTN · 2023-10-30

**Soundness:** 3 good
**Presentation:** 3 good
**Contribution:** 3 good
**Rating:** 6
**Confidence:** 5

**Summary:**

In this paper, the authors a simple yet effective long-tail learning framework namely PEL. Based on pretrained CLIP-ViT-B, PEL first introduces AdaptFormer and an individual cosine classifier to conduct parameter-efficient tuning on long-tailed data. To reduce the training difficulty, the authors also propose semantic-aware initialization, which leverages text embeddings from "a photo of [classname]" to initialize weights in the classifier. The proposed PEL achieves state-of-the-art performance on multiple long-tailed classification benchmarks.

**Strengths:**

1. The proposed efficient tunning method is effective, and the proposed semantic-aware initialization is intuitive and also effective.

2. The experimental results are promising, and the ablation studies are extensive.

3. This paper is well-written and easy to follow.

**Weaknesses:**

The main concern is also comes from semantic-aware initialization. Intuitively, only vision-language pretrained models (e.g., CLIP-pretrained models) support this kind of initialization. However, vision-only pretrained models cannot benefit from this initialization method. Though as shown in the code, using mean features in training set is alternative, the quality of tail classes is still not reliable like that of head classes. The authors could further discuss different initialization methods for different pretrained models (scenarios).

**Questions:**

See weakness section.

---

> ### Author Response · Authors · 2023-11-14
>
> We are grateful for the valuable reviews, and for the positive comments that the experimental results are *promising* and the paper is *well-written*. We will address the concerns below.
>
> **Concern #1: Initialization for vision-only pre-trained models.**
>
> **Response:** When employing vision-only pre-trained models, utilizing class mean features is a viable choice. As delineated in Table 5 in the main paper, this approach yields significant performance improvements. It surpasses most existing methods by comparing the results in Tables 1&2.
>
> Furthermore, we have tried linear probing with re-weighted (RW) or logit-adjusted (LA) loss and report the results below. The results are better than random initialization but are inferior to class mean features.
>
> It remains an intriguing challenge how to initialize the classifier for visual-only models with limited data. Anyway, employing class mean features proves to be an intuitive and effective approach at present.
>
> - ImageNet-LT
>
>   |                       | Overall  | Many     | Medium   | Few      |
>   | --------------------- | -------- | -------- | -------- | -------- |
>   | Random initialization | 76.1     | 80.8     | 75.9     | 63.2     |
>   | Linear probing w/ RW  | 77.1     | **81.8** | 76.4     | 66.6     |
>   | Linear probing w/ LA  | 77.2     | 81.3     | 76.4     | 68.4     |
>   | Class mean features   | **77.5** | 81.3     | **76.8** | **69.4** |
>
> - Places-LT
>
>   |                       | Overall  | Many     | Medium   | Few      |
>   | --------------------- | -------- | -------- | -------- | -------- |
>   | Random initialization | 49.6     | 51.2     | **52.1** | 41.0     |
>   | Linear probing w/ RW  | 49.9     | 51.2     | 51.0     | 45.0     |
>   | Linear probing w/ LA  | 50.2     | 51.3     | 51.1     | 46.2     |
>   | Class mean features   | **51.3** | **51.6** | **52.1** | **48.8** |

---

### Official Review · Reviewer_BCXb · 2023-10-31

**Soundness:** 3 good
**Presentation:** 3 good
**Contribution:** 2 fair
**Rating:** 5
**Confidence:** 5

**Summary:**

This paper proposes a fine-tuning method PEL that efficiently adapts CLIP to long-tailed recognition tasks, within typically 20 epochs. One main motivation is that both full and classifier fine-tuning suffer from overfitting, with decreased performance on tail classes. To address overfitting, class imbalance and large cost during finetuning, PEF features 4 components: 1) parameter-efficient fine-tuning (PEFT) to mitigate overfitting through optimizing only a small amount of parameters on top of a frozen backbone, 2) logit-adjusted (LA) loss for training a cosine classifier to address class imbalance, 3) semantic-aware initialization to speedup convergence, 4) test-time ensembling to further improve generalization. SOTA performance is obtained on four long-tailed datasets while the compute cost remains low.

**Strengths:**

- Table 8 provides nice benchmarking of the overfitting issue of popular finetuning methods, which acts as a strong motivation of using PEFT to mitigate overfitting.
- The four components of the proposed PEL method (listed above) make sense, and their combination does lead to a good perf-cost tradeoff, as shown in the many experiments.

**Weaknesses:**

- None of the 4 components in PEL is new, and the big featuring of existing techniques seems a bit ad-hoc, which raises novelty concerns.
- Each of the 4 components is ablated separately --- Tables 1-3 validate the need of test time ensembling, while Figs 9-10 show that both PEFT and semantic-aware initialization are helpful. Tables 9-10 respectively show that the adopted classifier and loss function are the right choice. However, there's no clue about which of the 4 existing techniques play a key role, and which is more like icing on the cake. To tell which components are more important, a systematic ablation is required but is missing in the current paper, e.g., baseline vs baseline+component 1 vs baseline+components 1&2 ... vs baseline+components 1&2&3&4.

**Questions:**

- Semantic-aware initialization is claimed to accelerate convergence by initializing the classifier weights with text prompt features. What if the class is OOD/unseen for CLIP? Will such class prompt features still be a good initialization?
- In Table 10, it seems that different loss functions are all about the tradeoff between tail and head class performance. For example, the adopted LA loss, despite achieving the best performance on Median/Few splits, is suboptimal on the Many splits. Any comments?

---

> ### Author Response · Authors · 2023-11-14
>
> We appreciate the reviewer for the thoughtful reviews and for the encouraging comments such as *nice benchmarking* and *strong motivation*. Following are our responses to the concerns.
>
> **Concern #1 (Weakness #1): The novelty of PEL.**
>
> **Response:** Since it is the first work that systematically studies the parameter-efficient adaptation of pre-trained models on long-tailed tasks, we aim to **reveal fundamental insights** and **inspire future research**.
>
> 1) Full Fine-tuning fails in long-tailed learning, and PEFT emerges as a superior alternative. This contradicts common sense on balanced datasets, where PEFT typically strives to match the performance of full fine-tuning [1-5].
> 2) We discover that most elaborately designed losses fail when applied to CLIP, except Logit-Adjusted loss (explained in Appendix B). This discovery motivates future research to design more suitable losses.
> 3) We point out the importance and advantages of classifier initialization, particularly for tail classes. This also inspires more advanced research in the future.
> 4) We first propose to employ TTE to remedy the inherent limitations of Vision Transformer. As a remedial measure, TTE is intuitive and effective (explained in Appendix C).
>
> Furthermore, PEL is a general framework, with all of the components efficient and scalable. We believe our findings and efforts can contribute to the long-tailed learning community.
>
>
>
> [1] BitFit: Simple Parameter-efficient Fine-tuning for Transformer-based Masked Language-models.
>
> [2] Visual Prompt Tuning.
>
> [3] Parameter-Efficient Transfer Learning for NLP.
>
> [4] LoRA: Low-rank adaptation of large language models
>
> [5] AdaptFormer: Adapting Vision Transformers for Scalable Visual Recognition.
>
>
>
> **Concern #2 (Weakness #2): A systematic ablation of components.**
>
> **Response:** Thanks for your valuable comments. We ablate the components in PEL separately, including 1) the **PEFT** module, 2) the Logit-Adjusted (**LA**) loss, 3) Semantic-Aware Initialization (abbreviated as **SAI**), and 4) Test-Time Ensembling (**TTE**). The results are reported below.
>
> The results demonstrate the effectiveness of each component. Specifically, 1) PEFT enhances performance on both head and tail classes; 2) without the LA loss, predictions tend to be biased to head classes; 3) Semantic-Aware Initialization (SAI) consistently improves the generalization, particularly on tail classes; 4) Test-Time Ensembling further boost the performance across both head and tail classes.
>
> - Ablation study on ImageNet-LT. The baseline involves learning a classifier using CE loss.
>
>   | PEFT         | LA           | SAI          | TTE          | Overall  | Many     | Medium   | Few      |
>   | ------------ | ------------ | ------------ | ------------ | -------- | -------- | -------- | -------- |
>   |              |              |              |              | 60.9     | 82.6     | 56.7     | 13.8     |
>   | $\checkmark$ |              |              |              | 68.9     | **84.7** | 66.3     | 33.7     |
>   | $\checkmark$ | $\checkmark$ |              |              | 74.9     | 79.7     | 74.7     | 61.7     |
>   | $\checkmark$ | $\checkmark$ | $\checkmark$ |              | 77.0     | 80.2     | 76.1     | 71.5     |
>   | $\checkmark$ | $\checkmark$ | $\checkmark$ | $\checkmark$ | **78.3** | 81.3     | **77.4** | **73.4** |
>
> - Ablation study on Places-LT. The baseline involves learning a classifier using CE loss.
>
>   | PEFT         | LA           | SAI          | TTE          | Overall  | Many     | Medium   | Few      |
>   | ------------ | ------------ | ------------ | ------------ | -------- | -------- | -------- | -------- |
>   |              |              |              |              | 34.3     | 53.6     | 30.5     | 7.5      |
>   | $\checkmark$ |              |              |              | 38.9     | **55.6** | 35.3     | 16.4     |
>   | $\checkmark$ | $\checkmark$ |              |              | 48.7     | 50.6     | 50.8     | 40.5     |
>   | $\checkmark$ | $\checkmark$ | $\checkmark$ |              | 51.5     | 51.3     | 52.2     | 50.5     |
>   | $\checkmark$ | $\checkmark$ | $\checkmark$ | $\checkmark$ | **52.2** | 51.7     | **53.1** | **50.9** |

---

> ### Author Response · Authors · 2023-11-14
>
> **Concern #3 (Question #1): What if the class is OOD/unseen for CLIP? Will such class prompt features still be a good initialization?**
>
> **Response:** We posit that CLIP has seen sufficient language corpus, considering its pre-training on web-scale datasets. However, it is noteworthy that CLIP may fail to recognize specific class names. This probably stems from its limited vocabulary size (CLIP contains approximately 49K of vocabulary) or encountering uncommon or novel concepts. In this case, the initialization may regress to random initialization.
>
> One solution is to design some descriptions for this class, which can be crafted manually or generated using large language models. We follow [6] to generate the descriptions. Following are some examples:
>
> | Class    | Descriptions                                                 |
> | -------- | ------------------------------------------------------------ |
> | Hen      | a chicken; red, brown, or white feathers; a small body; a small head; a beak; two legs; two wings; a tail |
> | Trombone | a brass musical instrument; long, coiled tubing; a mouthpiece; a slide; valves or keys; a bell |
>
> We conduct experiments on ImageNet-LT by assuming that the class names are unseen. We solely use the descriptions for initialization. The results shown below demonstrate that the descriptions can enhance generalizations, particularly for tail classes.
>
> |                       | Overall | Many | Medium | Few  |
> | --------------------- | ------- | ---- | ------ | ---- |
> | Random initialization | 76.1    | 80.8 | 75.9   | 63.2 |
> | Class descriptions    | 77.4    | 81.3 | 76.9   | 68.2 |
> | Class names           | 78.3    | 81.3 | 77.4   | 73.4 |
>
>
>
> **Concern #4 (Question #2): Different loss functions are all about the tradeoff between tail and head class performance.**
>
> **Response:** The phenomenon you highlighted does indeed exist. In the context of long-tailed learning, there exists a competition for decision boundaries between head and tail classes. By optimizing re-balancing losses, the model tends to predict samples as tail classes, leading to a performance deterioration of head classes, which is called the *seesaw dilemma*.
>
> Most recently, some works have proposed to address this issue through multi-objective optimization [7] [8]. Although the solution may not be fully mature, the topic remains intriguing and warrants exploration in the future.
>
>
>
> [6] Visual classification via description from large language models.
>
> [7] Long-Tailed Learning as Multi-Objective Optimization. In https://arxiv.org/abs/2310.20490.
>
> [8] Pareto Deep Long-Tailed Recognition: A Conflict-Averse Solution. In https://openreview.net/forum?id=b66P1u0k15.

---

> ### Comment · Reviewer_BCXb · 2023-11-21
> **Response to Author Feedback**
>
> Thanks for your detailed feedback and new ablations. Below are my follow-up comments.
>
> > Since it is the first work that systematically studies the parameter-efficient adaptation of pre-trained models on long-tailed tasks, we aim to reveal fundamental insights and inspire future research.
>
> At first sight, this paper seems a "performance" one by stacking up different available techniques to obtain SOTA results, which raises novelty concerns. Now the authors clarify that the paper is more about empirical evaluations of possible techniques for the same goal of ``parameter-efficient finetuning for long-tailed tasks''. Then it seems to me that the paper needs a major rewriting. More focus should be put on the comparisons and findings/insights from different threads of investigations, rather than on SOTA performance or any technical contributions. At least the authors need to tone down on the performance and technical contribution sides. The paper in its current form (e.g., title, abstract, the way it describes PEL, and the whole message) is misleading, and it seems to need a lot of time to be publication ready, which is concerning. Another lingering concern is the exploration of 4 independent techniques without good motivation of why they are selected (but not others) and/or enough analysis about the synergy between them.
>
> > The results shown below demonstrate that the descriptions can enhance generalizations, particularly for tail classes.
>
> The results seem to suggest that using class descriptions, despite being better than random initialization, is still inferior to using simple class names (lagging behind on the "few" group by a large margin). Am I interpreting the results right?

---

> ### Author Response · Authors · 2023-11-21
>
> Dear Reviewer,
>
> We sincerely appreciate your follow-up comments. We would like to provide further clarification on the concerns below.
>
> **Concern: At first sight, this paper seems a "performance" one by stacking up different available techniques to obtain SOTA results, which raises novelty concerns. Now the authors clarify that the paper is more about empirical evaluations of possible techniques for the same goal of "parameter-efficient finetuning for long-tailed tasks''.**
>
> **Response:** There may be some misunderstandings. The primary objective of this paper is to reveal the limitations of conventional fine-tuning methods and to propose a new framework for long-tailed recognition. Therefore, **the main contributions of this paper lie in the proposed framework and the state-of-the-art performance it achieves.**
>
> Moreover, it is essential to clarify that **our work is not about empirical evaluations**. The response to Concern#1 is not our motivations; instead, it summarizes the novel discoveries emerging from our works. These discoveries can be viewed as **broader impacts**, and we hope these discoveries can inspire future research.
>
>
>
> **Concern: The exploration of 4 independent techniques without good motivation of why they are selected (but not others) and/or enough analysis about the synergy between them.**
>
> **Response:** We understand your concern. However, we have to clarify that these 4 modules are all with sufficient motivation.
>
> - **Motivation for PEFT**: We have discussed the limitation of 1) zero-shot CLIP, 2) full fine-tuning, 3) classifier fine-tuning in Section 3.2, as well as the limitation of 4) partial fine-tuning in Paragraph *Partial fine-tuning vs. parameter-efficient fine-tuning* in Appendix A. Therefore, the question arises: How to adapt the pre-trained model to downstream long-tailed tasks? Adopting Parameter-efficient Fine-tuning (PEFT) emerges as an intuitive, economical, and effective approach.
>
>   Among various PEFT methods, we employ AdaptFormer due to its state-of-the-art performance. Additionally, we explain why it helps in Paragraph *Effects of the AdaptFormer module on different layers* in Appendix A.
>
> - **Motivation for LA loss**: It is widely recognized that employing re-balancing loss is a common method to alleviate the long-tailed problem. Additionally, we have theoretically explained that LA loss is more generalizable across different backbone models. Furthermore, we conduct experiments in Paragraph *Comparison of different losses* in Appendix A to demonstrate its superiority over other elaborately designed losses.
>
> - **Motivation for Semantic-aware initialization**: As we explained in Section 3.3, semantic-aware initialization is a simple approach to leverage the semantic relationships, which can effectively make up for the data scarcity issues on the tail classes. The results in Table 5 demonstrate its ability to enhance tail-class performance. We also demonstrate its effectiveness on fast convergence in Figure 5.
>
> - **Motivation for TTE**: As we explained in Section 3.3 and Appendix C, TTE is specially designed to compensate for the limitation of Vision Transformer, and can enhance the generalization on both head and tail classes.
>
> All these modules are specially designed for long-tailed learning with pre-trained models and they have been proven effective in enhancing generalization, particularly on tail classes.
>
>
>
> **Concern: The results seem to suggest that using class descriptions, despite being better than random initialization, is still inferior to using simple class names (lagging behind on the "few" group by a large margin).**
>
> **Response:** Thanks for your concern. Your understanding is correct since the class descriptions do not contain the class names at all. When employing these descriptions for zero-shot predictions, the performance is notably inferior compared to using class names, as illustrated in the table below.
>
> - Zero-shot CLIP performance with class descriptions or class names
>
>   |                    | Overall | Many | Medium | Few  |
>   | ------------------ | ------- | ---- | ------ | ---- |
>   | class descriptions | 25.8    | 28.1 | 24.7   | 23.3 |
>   | class names        | 68.3    | 69.2 | 67.6   | 67.7 |
>
> Nevertheless, leveraging class descriptions remains **a viable alternative approach when the class names are unachievable or OOD for CLIP**. We propose this in response to your concerns "What if the class is OOD/unseen for CLIP?" The results demonstrate that using class descriptions can effectively mitigate such challenges.

---

### Author Response · Authors · 2023-11-18
**We look forward to your further feedback**

Dear Reviewers,

We sincerely appreciate all of you for your time and effort. Your valuable comments are very helpful for improving our paper. In addition to the responses, **we have also updated our manuscript** in response to your concerns. The following are the details:

- To avoid misunderstanding, we update **Table 4** and Paragraph *On parameter-efficient fine-tuning methods*.
- For systematical ablation studies, we add **Table 11** in Appendix A, and discuss the results in Paragraph *Ablation study of components*.
- For adopting ResNet as the backbone, we add **Tables 12 and 13** in Appendix A, and discuss the results in Paragraph *PEL with ResNet as the backbone*.
- For the comparison of different augmentation methods, we add **Table 14** in Appendix C, as well as some discussions in Appendix C.
- To discuss textual prompts for semantic-aware initialization, we add a new section **Appendix D**, where we discuss applying different templates or using class descriptions as an alternative.
- For the time cost of different PEFT methods, we add **Table 16** in Appendix E, as well as some discussions in Appendix E.

Moreover, it is important for us to know whether our responses have addressed your concerns, and we look forward to receiving your further feedback.

Best Regards,

Authors

---

### Meta-Review · Area_Chair_b7yz · 2023-12-06

**Metareview:**

This paper proposes an efficient fine-tuning of pre-trained vision-language models on long-tail distribution. Based on observations that zero-shot learning or simple fine-tuning suffer from either distribution shift or severe over-fitting, the authors proposed to incorporate combinations of well-known techniques in the literature, such as parameter-efficient fine-tuning, logit adjustment, and test-time ensembling. The authors also showed that further improvement can be made by initializing the classifier weights using the text features in the vision language model.

The paper received mixed ratings of three borderline reject and one borderline accept. The critical concerns raised by the reviewers were about (1) limited technical novelty, since most of the components in the proposed method are simple integration of well-known techniques, (2) the positioning of the paper is misleading, since the paper mostly focuses on revealing the empirical observations of fine-tuning on long-tail recognition task while some arguments are highlighting the technical contributions, and (3) some missing analysis in the experiments. Although additional experiment results presented during the rebuttal have adequately addressed some of the reviewers’ concerns, all reviewers agreed that the technical contribution is still marginal, and repositioning of the paper is necessary.

After carefully reading the paper, reviews, and rebuttal, AC thinks that the paper provides meaningful empirical findings on exploiting pre-trained vision-language models on long-tail recognition tasks. However, apart from the marginal technical novelty, AC also agrees with the reviewers that the insights from the paper, based on the current draft, are not strong enough to offset the limited novelty. It has been widely observed in the literature that simple fine-tuning of pre-trained models to few-shot labels, even with the classifier-only fine-tuning,  leads to overfitting and sacrificing the generalization performance. In such a regime, parameter-efficient fine-tuning has also been considered as a superior alternative to full fine-tuning, while the advantage diminishes as the size of labeled data increases. It is thus natural to observe similar trends in long-tail recognition tasks over the head and tail classes, as well as observe improvement by applying logit adjustment to the classifier for handling imbalanced classes. Additionally, AC thinks that some of the baselines could be underestimated in the paper and suggests authors consider (1) classifier-only fine-tuning with logit adjustment as baseline, since the improvement by LA alone seems already significant based on ablation study presented in the rebuttal, and (2) more advanced choice of promptings in the zero-shot baseline, since the current approach employs simple class name for the prompting yet its performance is highly sensitive to the choice of prompts.

Given these reasons, AC believes that the current paper is not on par with the ICLR standard and hence recommends rejection this time.

**Justification For Why Not Higher Score:**

N/A

**Justification For Why Not Lower Score:**

N/A

---

### Decision · Program_Chairs · 2024-01-16

Reject